# A clinically applicable deep-learning model for detecting intracranial aneurysm in computed tomography angiography images

Zhao Shi [1,10], Chongchang Miao[2,10], U. Joseph Schoepf[3], Rock H. Savage[3], Danielle M. Dargis[3], Chengwei Pan [4,10], Xue Chai[5,10], Xiu Li Li[6,10], Shuang Xia[7,10], Xin Zhang[8,10], Yan Gu[2], Yonggang Zhang[2], Bin Hu[1], Wenda Xu[1], Changsheng Zhou[1], Song Luo[1], Hao Wang[6], Li Mao [6], Kongming Liang[6], Lili Wen[8], Longjiang Zhou[8], Yizhou Yu [6], Guang Ming Lu [1✉] & Long Jiang Zhang [1,9✉]

Intracranial aneurysm is a common life-threatening disease. Computed tomography angiography is recommended as the standard diagnosis tool; yet, interpretation can be time-consuming and challenging. We present a specific deep-learning-based model trained on 1,177 digital subtraction angiography verified bone-removal computed tomography angiography cases. The model has good tolerance to image quality and is tested with different manufacturers. Simulated real-world studies are conducted in consecutive internal and external cohorts, in which it achieves an improved patient-level sensitivity and lesion-level sensitivity compared to that of radiologists and expert neurosurgeons. A specific cohort of suspected acute ischemic stroke is employed and it is found that 99.0% predicted-negative cases can be trusted with high confidence, leading to a potential reduction in human workload. A prospective study is warranted to determine whether the algorithm could improve patients' care in comparison to clinicians' assessment.

[1] Department of Diagnostic Radiology, Jinling Hospital, Medical School of Nanjing University, Nanjing, Jiangsu 210002, P.R. China. [2] Department of Radiology, Lianyungang First People's Hospital, Lianyungang, Jiangsu 222002, P.R. China. [3] Division of Cardiovascular Imaging, Department of Radiology and Radiological Science, Medical University of South Carolina, Charleston, SC, USA. [4] Computer Science Department, School of EECS, Peking University, Beijing 100089, P.R. China. [5] Department of Radiology, Affiliated Nanjing Brain Hospital, Nanjing Medical University, Nanjing, Jiangsu 210002, P.R. China. [6] DeepWise AI lab., Beijing 100089, P.R. China. [7] Department of Radiology, Tianjin First Central Hospital, Tianjin 300192, P.R. China. [8] Department of Neurosurgery, Jinling Hospital, Medical School of Nanjing University, Nanjing, Jiangsu 210002, P.R. China. [9] Department of Diagnostic Radiology, Jinling Hospital, Sothern Medical University, Nanjing, Jiangsu 210002, P.R. China. [10]These authors contributed equally: Zhao Shi, Chongchang Miao, Chenwei Pan, Xue Chai, Xiuli Li, Shuang Xia, Xin Zhang. ✉email: cjr.luguangming@vip.163.com; kevinzhlj@163.com

Intracranial aneurysms (IAs) are relatively common life-threatening diseases with a prevalence of 3.2% in the general population[1] and account for 85% in the spontaneous subarachnoid hemorrhage (SAH) patients[2]. IAs are increasingly being detected owing to the widespread application of advanced imaging techniques. Although aneurysmal SAH accounts for 5–10% of all strokes in the United States[3], it may cause significantly high mortality[4], and the survivors may suffer from long-term neuropsychological effects and decreased quality of life[5]. Early diagnosis of underlying IAs can both influence clinical management and guide prognosis in intracerebral hemorrhage patients[6,7]. For patients with spontaneous SAH, timely and accurate identification of IAs is critical for immediate intervention or surgical management, whereas for patients without IAs, reliable exclusion of IAs is also important for specialized management[8].

Computed tomography angiography (CTA) is a noninvasive, convenient, and reliable modality to detect IAs[9]. American Heart Association/American Stroke Association guidelines have recommended CTA as a useful tool for detection and follow-up of unruptured IAs (Class I; Level B)[10] and the work-up of aneurysmal SAH (Class IIb; Level C)[4]. However, CTA interpretation is time-consuming and requires subspecialty training. The existing challenges also include inter-observer variability and high false-negative (FN) rates[11–13]. The diagnostic accuracy is dependent on several factors including aneurysm size, diversity of technological specifications (16- versus 64-detector rows), image acquisition protocols, image quality, image postprocessing algorithms, and variations in radiologists' level of experience. These factors result in a mean sensitivity in the range of 28–97.8% in detecting IAs[9,13,14]. The recently published guideline of acute ischemic stroke (AIS) has also strongly recommended CTA use (Class I; level A) in selecting candidates for mechanical thrombectomy after illness onset[15]. Consequently, the workload of radiologists to detect or exclude IAs is rapidly increasing in the non-SAH setting, where excluding IAs on CTAs remains a challenging task. Given all the preconditions mentioned above, it is a timely and urgent need to have high-performance computer assisted diagnosis (CAD) tools to add in detection, increase efficiency, and reduce disagreement among observers which may potentially improve clinical care of patients.

MR angiography (MRA) or CTA-based CAD programs have been devised to automatically detect IAs. The conventional-style CAD systems were based on pre-supplied characteristics or imaging features, such as vessel curvature, thresholding, or a region-growing algorithm[16,17]. In addition, their performances in the real-world and their generalization have not been fully investigated. Nowadays, deep learning (DL) has shown significant potential in accurately detecting lesions on medical imaging and has reached, or perhaps surpassed, an expert-level of diagnosis[18–20,21]. DL is a machine learning technique that directly learns the most predictive features from a large data set of labeled images[20,21]. Explorations on DL combined with MRA have reported decent results for IAs detection[22,23]. CTA-based CAD systems for automatically detecting IAs have been rarely reported, and only two recently published studies can be found in this field, to the best of our knowledge[24,25]. Notably, these models were trained with a small sample size, without an outside reference standard, and were not tested in different scenarios; thus, they were not adequate to be applied in real-world clinical settings and may fall into the "Artificial intelligence (AI) chasm", which can be described as divide between developing a scientifically sound algorithm and its use in any meaningful real-world applications[26]. It is necessary and imperative to develop a robust and reliable AI tool for IAs in the clinical real-world setting. The experiments presented in this article will endeavor to resolve these problems.

To properly address the shortcomings of current computational approaches in the context of IAs detection and to enable clinical deployment, training and validation of models on large-scale data sets representative of the wide variability of cases encountered in the routine clinic is required. Therefore, we collected 1177 digital subtraction head bone-removal CTA images, which were based on a section-by-section subtraction to subtract nonenhanced from enhanced CT data to facilitate the diagnosis of aneurysms[27], with/without SAH to derive a specific model for automated detection of IAs. Furthermore, we aimed to ascertain the capability and generalizability of the model using temporally or spatially independent real-world CTA images from four hospitals and to test the model in a simulated real-world routine clinical setting. In order to guarantee the ground truth of IAs, especially small ones, we only enrolled patients who underwent cerebral digital subtraction angiography (DSA), which is the gold standard for diagnosing IAs, to verify the results of CTA in the training data set.

In this work, the specific model was an end-to-end 3D convolutional neural network (CNN) segmentation model (Fig. 1). First, an encoder–decoder architecture was used for smooth and gradual transitions from original images to segmentation mask (Supplementary Fig. 1). Second, residual blocks[28] were adopted to allow for stable training for increasing depth of the network. Third, a dual attention[29] block was embedded to learn long-range contextual information to get more reliable feature representations. The comprehensive analysis of the proposed model on the effects of different factors and in real-world clinical scenarios is conducted in eight cohorts in total (Table 1). The results demonstrate that the model has a good tolerance to image quality and different manufacturers had a slight impact on the performance. The model has an improved patient-level sensitivity and lesion-level sensitivity compared to that of radiologists and expert neurosurgeons. Besides, we note that 99.0% predicted-negative cases in AIS setting can be trusted with high confidence, which leads to a potential reduction in human workload.

## Results

**Model development and primary validation.** There were 16,975 consecutive digital subtraction head CTAs and 7035 cerebral DSAs performed between 1 June 2009 and 31 March 2017 in Jinling Hospital, among which 1875 patients underwent both CTA and DSA. After quality-control evaluation and image review, the final data set (Internal cohort 1) consisted of 1177 cases (316 slices/case): 869 patients with 1099 aneurysms and 308 non-aneurysm controls, including 257 patients without abnormal findings and 51 patients with intracranial artery stenosis. The cohort was split into training/tuning/testing sets. The training set included 927 cases (744 cases with IAs and 183 non-aneurysm controls); the tuning set consisted of 100 cases (50 cases with IAs and 50 controls); the testing set had 150 cases with 50% of cases having IAs (94 aneurysms totally) (Supplementary Fig. 2).

The DL algorithm was developed on the training set. After the entire training procedure (see Methods), the model with the best lesion-level sensitivity on the tuning set was cherry-picked. Furthermore, the patient-level sensitivity and specificity of the model under different false positives per case (FPs/case) were analyzed (Supplementary Fig. 3), and the comprehensive optimal threshold was determined to make the FPs/case to be 0.29/case so that the model reached a high sensitivity of 97.3% with a moderate specificity of 74.7% on the testing set. The lesion-level sensitivity was 95.6% with a Dice ratio of 0.75. In all, four aneurysms were missed in four patients, including two patients with multiple IAs (one aneurysm was missed for each patient). All of the missed aneurysms were small (<5 mm) and three were

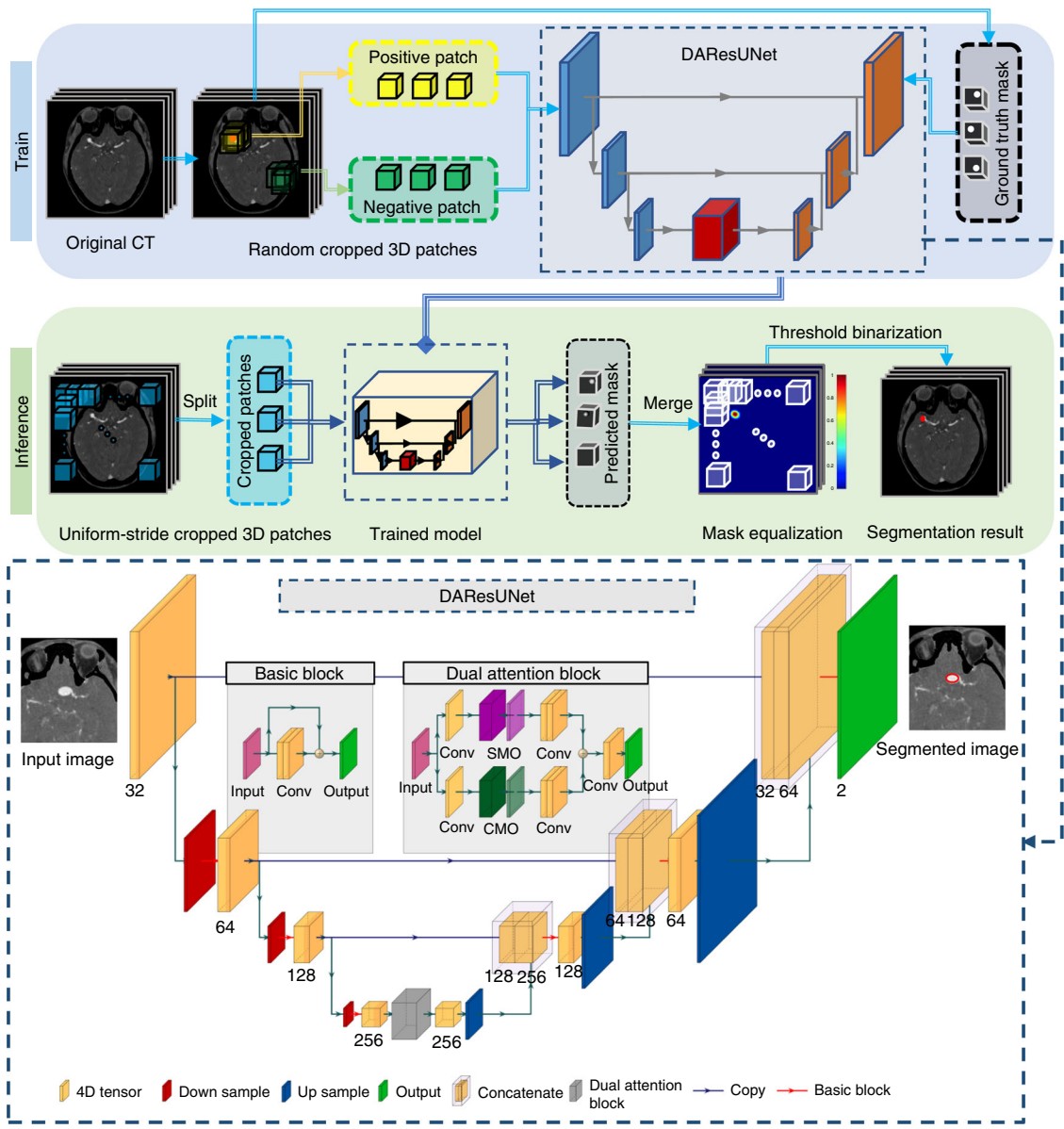

**Fig. 1 Overview of the proposed DL model presented in this study. a** Training stage: we used 3D patches randomly sampled from the digital subtraction bone-removal CTA scans to train the network. **b** Inference stage: uniform-stride sampling was used and then the prediction of those samples was merged to obtain the final prediction of the whole volume. **c** Illustration of the architecture of the end-to-end aneurysm prediction model. The proposed segmentation model had a similar encoder–decoder architecture as U-Net[30], and residual blocks[28] and a dual attention block[29] were used to improve the performance of the network. *IA* intracranial aneurysm, *SMO* spatial matrix operation, *CMO* channel matrix operation.

tiny IAs (<3 mm), which result in a 100% lesion-level sensitivity for aneurysms ≥5 mm and 98.6% for those ≥3 mm. Two of the missed aneurysms were located in the cerebellar artery (CA, one in the right posterior inferior CA and the other in the left superior CA), 1 was located in internal carotid artery (ICA) and posterior communication artery (PCoA). For the remaining locations, the model demonstrated 100% lesion-level sensitivity. The model took a mean of 17.6 s (95% CI: 17.2–18.0 s) to process an examination and output its segmentation map.

A completely independent internal validation data set (Internal cohort 2) was applied to test the model, which contained 245 cases (145 aneurysms in 108 patients) who had undergone both head CTA and DSA from 1 April 2017 to 31 December 2017 in Jinling Hospital (Table 1 and Supplementary Table 1). The model reached an accuracy, patient-level sensitivity, and specificity of 88.6%, 94.4%, and 83.9%, respectively, with a lesion-level

sensitivity of 84.1% and FPs/case of 0.26 (Table 2). In all, six patients with IAs were misdiagnosed, and 23 aneurysms in 20 patients were missed. The model had a high lesion-level sensitivity for aneurysms located in the anterior communication artery (ACoA, 100%), anterior cerebral artery (ACA, 100%), vertebral basilar artery (VBA, 100%) and PCoA (87.9%), middle cerebral artery (MCA, 87.5%), whereas lower lesion-level sensitivity for those in the ICA (60.6%), posterior cerebral artery (PCA, 66.7%), and CA (66.7%). The most frequently missed aneurysms were tiny aneurysms (<3 mm, lesion-level sensitivity of 51.7%), but there was an improved lesion-level sensitivity of 75.0% for those ≥3 mm, 95.8% for those ≥5 mm, 100% for those ≥10 mm. Supplementary Fig. 4 demonstrates some examples of the correctly predicted findings.

Another completely independent external validation data set from Nanjing Brain Hospital (NBH cohort) from 1 January 2019

**Table 1 Overview of the baseline characteristics of the eight cohorts.**

| | Internal cohort 1, n = 1177 | Internal cohort 2, n = 245 | Internal cohort 3, n = 226 | Internal cohort 4, n = 374 | Internal cohort 5, n = 333 | NBH cohort, n = 211 | TJ cohort, n = 147 | LYG cohort, n = 316 |
|---|---|---|---|---|---|---|---|---|
| Patients with IAs, n (%) | 869 (73.8) | 108 (44.1) | 61 (27.0) | 53 (14.2) | 14 (4.2) | 39 (18.5) | 109 (74.1) | 60 (19.0) |
| Number of IAs, n | 1099 | 145 | 80 | 71 | 16 | 46 | 141 | 76 |
| Patients without IAs, n (%) | 308 (26.2) | 137 (55.9) | 165 (73.0) | 321 (85.8) | 319 (95.8) | 172 (81.5) | 38 (25.9) | 256 (81.0) |
| Male sex, n (%) | 311 (28.3) | 136 (55.5) | 134 (59.3) | 242 (64.7) | 224 (67.3) | 145 (68.7) | 71 (48.3) | 187 (59.2) |
| SAH, n (%) | 931 (79.1) | 108 (44.1) | 69 (30.5) | 28 (7.5) | 0 | 5 (2.4) | 64 (43.5) | 47 (14.9) |
| Number of patients with IAs, n (%) | 760 (81.6) | 87 (80.6) | 44 (63.8) | 10 (35.7) | 0 | 2 (40.0) | 62 (96.9) | 25 (53.2) |
| Non-SAH (IA %) | 246 (20.9) | 137 (54.9) | 157 (69.5) | 346 (92.5) | 333 (100) | 206 (97.6) | 83 (56.5) | 269 (85.1) |
| Number of patients with IAs, n (%) | 109 (44.3) | 24 (17.5) | 17 (10.8) | 43 (12.4) | 14 (4.2) | 37 (18.0) | 47 (56.6) | 35 (13.0) |
| age, years | 54 (46,62) | 59 (48,66) | 59 ±13 | 63.0 (50.0,70.0) | 66 (57,77) | 64 (56,71) | 60 ±13 | 65.5 (52.5,73.0) |
| age <70 years, n (%) | 1088 (92.4) | 209 (85.3) | 173 (76.5) | 273 (73.0) | 198 (59.5) | 150 (71.1) | 106 (72.1) | 223 (70.6) |
| Location | | | | | | | | |
| MCA, n (%) | 131 (11.9) | 25 (16.9) | 8 (10.0) | 13 (18.3) | 6 (37.5) | 3 (6.5) | 26 (18.4) | 13 (17.1) |
| ACoA, n (%) | 291 (26.5) | 29 (19.6) | 20 (25.0) | 8 (11.3) | 1 (6.3) | 5 (10.9) | 29 (20.6) | 10 (13.2) |
| ICA, n (%) | 207 (18.8) | 34 (23.0) | 19 (23.8) | 15 (21.1) | 5 (31.3) | 21 (45.7) | 27 (19.1) | 19 (25.0) |
| PCoA, n (%) | 322 (29.3) | 33 (22.3) | 25 (31.3) | 19 (26.8) | 3 (18.8) | 5 (10.9) | 23 (16.3) | 19 (25.0) |
| VBA, n (%) | 40 (3.6) | 7 (4.7) | 0 | 9 (12.7) | 1 (6.3) | 8 (17.4) | 23 (16.3) | 7 (9.2) |
| CA, n (%) | 43 (3.9) | 6 (4.1) | 2 (2.5) | 3 (4.2) | 0 | 0 | 2 (1.4) | 1 (1.3) |
| ACA, n (%) | 41 (3.7) | 11 (7.4) | 5 (6.3) | 4 (5.6) | 0 | 3 (6.5) | 6 (4.3) | 5 (6.6) |
| PCA, n (%) | 24 (2.3) | 3 (2.0) | 1 (1.3) | 0 (0) | 0 | 1 (2.2) | 5 (3.5) | 2 (2.6) |
| Size | | | | | | | | |
| <3, n (%) | 204 (18.6) | 32 (21.6) | 22 (27.5) | 22 (31.0) | 0 | 8 (17.4) | 18 (12.8) | 18 (23.7) |
| ≥3, <5, n (%) | 426 (38.8) | 46 (31.1) | 35 (43.8) | 25 (35.2) | 7 (43.8) | 17 (37.0) | 48 (34.0) | 30 (39.5) |
| ≥5, <10, n (%) | 401 (36.5) | 65 (43.9) | 18 (22.5) | 16 (22.5) | 6 (37.5) | 18 (39.1) | 64 (45.4) | 21 (27.6) |
| ≥10, n (%) | 68 (6.1) | 5 (3.4) | 5 (6.3) | 8 (11.3) | 3 (18.8) | 3 (6.5) | 11 (7.8) | 7 (9.2) |
| Size, mm | 4.3 (3.0,6.0) | 4.8 (3.3,6.3) | 4.2 (2.8,5.1) | 3.5 (2.9,6.5) | 5.1 (3.7,7.8) | 4.4 (3.4,6.7) | 5.3 (3.6,7.2) | 4.4 (3.2,6.2) |

ACA anterior cerebral artery, ACoA anterior communication artery, IA intracranial aneurysm, ICA internal carotid artery, CA cerebellar artery, MCA middle cerebral artery, PCA posterior cerebral artery, PCoA posterior communication artery, SAH subarachnoid hemorrhage, VBA vertebral basilar artery.

**Table 2 Performance of the model in all cohorts.**

| Cohort | Accuracy | Patient-level sensitivity | Specificity | PPV | NPV | Lesion-level sensitivity | FPs/case | Dice ratio |
|---|---|---|---|---|---|---|---|---|
| Internal cohort 1 | 86.0% (79.5%–90.7%) | 97.3% (90.8%–99.3%) | 74.7% (63.8%–83.1%) | 79.4% (70.0%–86.4%) | 96.6% (88.3%–99.1%) | 95.6% (89.1%–98.3%) | 0.29 | 0.75 |
| Internal cohort 2 | 88.6% (83.7%–92.1%) | 94.4% (87.8%–97.7%) | 83.9% (76.5%–89.5%) | 82.3% (74.1%–88.3%) | 95.0% (89.1%–98%) | 84.1% (76.9%–89.5%) | 0.26 | 0.65 |
| Internal cohort 3 | 83.6% (78%–88.1%) | 78.7% (66%–87.7%) | 85.5% (78.9%–90.3%) | 66.7% (54.5%–77.1%) | 91.6% (85.7%–95.2%) | 68.8% (57.3%–78.4%) | 0.19 | 0.52 |
| Internal cohort 4 | 85.8% (81.8%–89.1%) | 73.6% (59.4%–84.3%) | 87.9% (3.6%–91.1%) | 50.0% (39.2%–60.8%) | 95.3% (81.8%–89.1%) | 60.6% (48.2%–71.7%) | 0.20 | 0.45 |
| Internal cohort 5 | 89.2% (85.2%–92.2%) | 78.6% (48.8%–94.3%) | 89.7% (85.7%–92.7%) | 25.0% (13.7%–40.6%) | 99.0% (96.7%–99.7%) | 68.8% (41.5%–87.9%) | 0.13 | 0.47 |
| NBH cohort | 81.0% (75–86%) | 84.6% (68.8%–93.6%) | 80.2% (73.3%–85.7%) | 49.3% (37%–61.6%) | 95.8% (90.8%–98.3%) | 76.1% (62.1%–86.1%) | 0.27 | 0.53 |
| TJ cohort | 8% (66.9%–81.5%) | 76.1% (66.9%–83.6%) | 71.1% (53.9%–84.0%) | 88.3% (79.6%–93.7%) | 50.9% (37%–64.7%) | 73.0% (64.8–80%) | 0.44 | 0.45 |
| LYG cohort | 76.6% (71.4%–81.1%) | 85.0% (72.9%–92.5%) | 74.6% (68.7%–79.7%) | 44.0% (34.9%–53.5%) | 95.5% (91.4%–97.8%) | 78.9% (67.8%–87.1%) | 0.32 | 0.56 |

The data in parentheses are 95% confidence interval.
FPs false positives, NPV negative predictive value, PPV positive predictive value.

to 31 July 2019 was collected to test the model's generalizability and robustness, which contained 211 cases with both head CTA and DSA. Of those cases, 39 patients had 46 IAs. The model reached an accuracy, patient-level sensitivity, and specificity of 81.0%, 84.6%, and 80.2%, respectively, with a lesion-level sensitivity of 76.1% and FPs/cases of 0.27 (Table 2). In all, 11 aneurysms from 11 patients (including five IAs in five patients with multiple IAs) were missed. The model had 100% lesion-level sensitivity for aneurysms located in the MCA, ACoA, ACA and PCA, and 80.0%, 66.7%, and 62.5% for those in the PCoA, ICA, and VBA. Similarly, it had lower lesion-level sensitivity for tiny aneurysms (37.5%), with a lesion-level sensitivity of 84.2% for those ≥3 mm, 90.5% for those ≥5 mm, 100% for those ≥10 mm. There was no significant difference between the patient-level, lesion-level sensitivity and specificity in the internal and external validation cohorts ($p = 0.114$, 0.239, and 0.400, respectively).

**Comprehensive analysis of the performance of the model.** In the process of model development, we found 31 occult cases that were defined as CTA-negative but DSA-positive aneurysms[8], especially for tiny aneurysms or those located in the ophthalmic artery segments (Supplementary Fig. 5). These cases were excluded in the internal cohort 1 and the validation cohorts because they were impossible to be annotated in CTA source images. We wondered whether these cases could be detected by the model given the strength of DL[18–20,21]. There are 39 aneurysms totally regarded as occult aneurysms (mean size: 2.0 (1.0, 3.0) mm, range: 1.0–4.0 mm). Most of these aneurysms were located in the ICA (51.3%, 20/39), especially in the ophthalmic artery segments (60.0%, 12/20). Our model detected five occult aneurysms with a mean size of 2.7 mm in five patients, among which two were located in the ICA, two in the CA, and one in the ACA.

Image quality affects the diagnostic performance of CTA, especially for small aneurysms. Herein, we analyzed the level of tolerance of our model to the image quality of head CTA. CTA image quality was rated on a four-point scale[31], which is based on the degree of noise, vessel sharpness, and overall quality (Supplementary Fig. 6). Further analysis was performed on another internal validation data set (Internal cohort 3) including 226 patients. In total, 61 patients had 80 aneurysms. There were 11, 55, 97, and 63 cases for the image quality scores 1–4, respectively. The results demonstrated that the patient-level sensitivities were 66.7%, 93.3%, 75.0%, and 72.7% in the groups of score 1–4 with a lesion-level sensitivity of 66.7%, 84.2%, 62.8%, and 66.7%; the specificities were 87.5%, 90.0%, 83.1%, and 92.3%, respectively (Supplementary Fig. 7b/c). There was no significant difference among the four subgroups (all Bonferroni-corrected $p > 0.05$), which meant the model had a relatively high tolerance to image quality. However, another truth is that with the advance of CT techniques and optimized CTA protocols, poor quality images (score 1) are rare. We only collected 11 cases with poor image quality among all head CTA cases, which meant the model requires further validation.

Manufacturers can be a factor affecting the performance of the model, so consecutive patients with head CTA from the Tianjin First Central Hospital (TJ cohort) from 1 January 2013 to 31 December 2018 were included, in which three different manufacturers were used to generate images. CTA was acquired by GE Revolution in 30 patients (26 patients with IAs), by SIEMENS SOMATOM Definition or Definition Flash in 68 (63 patients with IAs), and by Toshiba Aquilion ONE in 49 (20 patients with IAs). The results demonstrated that the lesion-level sensitivity of SIEMENS was significantly higher of 89.3% than those of GE (62.5%, $p = 0.001$) and Toshiba (32.0%, $p < 0.001$). The patient-level sensitivities were 90.5%, 69.2%, and 40.0% for

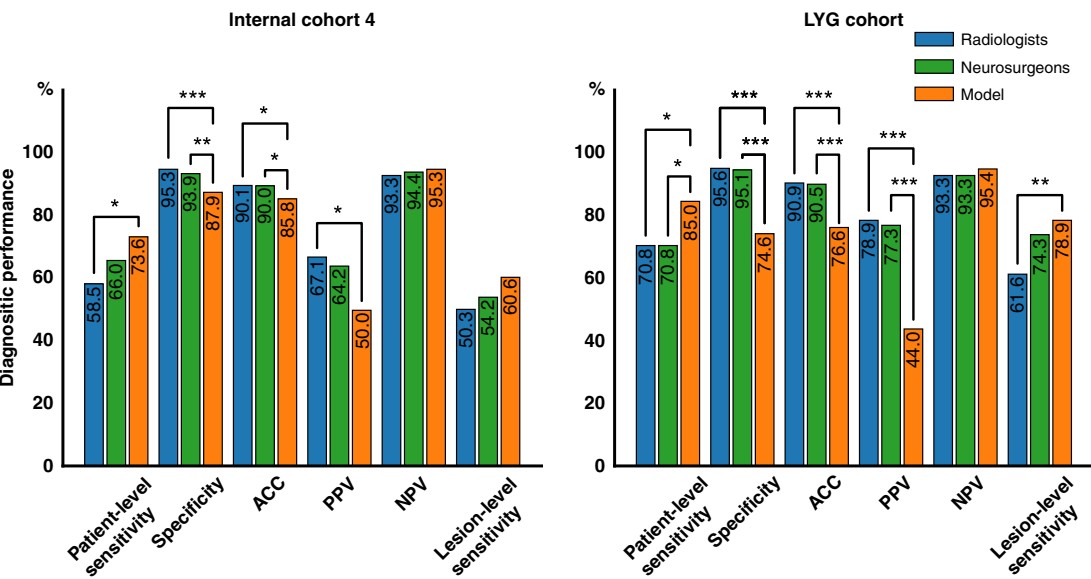

**Fig. 2 Comparison of the performance of the model and the radiologists/neurosurgeons in Internal cohort 4 and LYG cohort.** A two-sided Pearson's chi-squared test or Fisher exact test was used to evaluate the differences between the model and the radiologists and neurosurgeons. **a** Performance in Internal cohort 4. The model had higher patient-level sensitivity than that of the radiologists ($\chi^2 = 4.337$, $p = 0.037$) and comparative to neurosurgeons ($\chi^2 = 0.934$, $p = 0.334$). **b** Performance in external LYG cohort. Similar results can be found, i.e., that the model had a higher patient-level sensitivity than those of the radiologists ($\chi^2 = 5.219$, $p = 0.022$) and neurosurgeons ($\chi^2 = 4.347$, $p = 0.037$), and the specificity ($\chi^2 = 140.346$, $p < 0.001$ for the radiologists and $\chi^2 = 69.381$, $p < 0.001$ for the neurosurgeons), ACC ($\chi^2 = 55.784$, $p < 0.001$ for the radiologists and $\chi^2 = 33.652$, $p < 0.001$ for the neurosurgeons) and PPV ($\chi^2 = 49.458$, $p < 0.001$ for the radiologists and $\chi^2 = 26.137$, $p < 0.001$ for the neurosurgeons) were significantly lower for the model. *ACC* accuracy, *NPV* negative predictive value, *PPV* positive predictive value. *$0.01 \leq p < 0.05$; **$0.001 \leq p < 0.01$; ***$p < 0.001$.

SIEMENS, GE, and Toshiba. The specificity was comparable with 100% in GE and SIEMENS, whereas 58.6% for TOSHIBA without significant differences (Bonferroni-corrected $p > 0.05$) (Supplementary Fig. 7a/c).

**Radiology expert analysis of the error modes.** To uncover the underlying causes of misclassified cases in our developed model, we analyzed the cases in Internal cohorts 2, 3, NBH cohort, and TJ cohort. The misdiagnosed cases (6 in Internal cohort 2, 13 in Internal cohort 3, 6 in NBH cohort and 26 in TJ cohort) were found between the model diagnosis and the DSA findings (Supplementary Table 2a). Since that misdiagnosis would lead to serious consequences, we focused on these cases rather than the FPs. Several reasons can be attributed to misdiagnosed cases, including (1) tiny aneurysms (<3 mm); (2) Uncommon shape of aneurysms; (3) aneurysms located in uncommon locations (such as the posterior inferior CA); (4) inappropriate contrast agent protocols that result in poor artery enhancement or marked intracranial vein enhancement; (5) overshooting of bone subtraction in the ophthalmic artery segment; (6) other unexplained reasons (Supplementary Table 2b). We defined unexplained reasons as these aneurysms that were obvious for radiologists to identify but were missed by our model. The unexplained cases were mainly found in the TJ cohort and NBH cohort, which may be attributed to different manufacturers or CTA scan protocols (representative missed cases are presented in Supplementary Fig. 8).

**Clinical application in routine practice and comparison with radiologists and neurosurgeons.** In order to clearly understand the performance of the model against clinicians in the clinical setting, we designed an experiment to compare the performance of the model to those of six board-certified radiologists (two resident radiologists, F.X. and Y.X.; two attending radiologists, L.

W. and X.L.Z.; two assistant director radiologists, J.L. and Y.E.Z.) and two assistant director neurosurgeons (L.L.W. and L.J.Z.) in reading consecutive real-world cases with suspected IAs or other cerebral vascular disease that underwent head CTA, in one internal data set from 1 June 2019 to 31 July 2019 in Jinling Hospital (Internal cohort 4) and one external data set from 1 August 2018 to 30 September 2019 in Lianyungang First People's Hospital (LYG cohort). The prevalence of IAs is distinct in the general populations and SAH patients[1,2], which acts as an important clue for detecting IAs for clinicians. Therefore, we analyzed the performance of the model and clinicians in the entire group, SAH subgroup, and non-SAH subgroup, respectively.

For Internal cohort 4, the micro-averaged patient-level sensitivity and specificity were 58.5%, 95.3%; 66.7%, 95.4%; 56.6%, 95.3% for the radiologists in the entire group, SAH group and non-SAH group, respectively. The neurosurgeons had a patient-level sensitivity and specificity of 66.0%, 93.9%; 80.0%, 83.3%; and 62.8%, 94.6%. The radiologists had higher positive predictive value (PPV) in the SAH group than the non-SAH group ($p < 0.001$), and vice versa for negative predictive value (NPV) ($p < 0.001$), which demonstrated the additional value of SAH for diagnosis of IAs[9] (Supplementary Table 3). The micro-averaged lesion-level sensitivities were 50.3%, 54.8%, and 49.1% for radiologists and 54.2%, 64.3%, and 51.8%, respectively, for neurosurgeons in the three groups. For the model, it had higher patient-level sensitivity (73.6% ($p = 0.037$ and 0.334, compared with radiologists and neurosurgeons, respectively), 80.0% ($p = 0.636$ and $p > 0.999$) and 72.1% ($p = 0.056$ and 0.293)) and NPV (95.3% ($p = 0.370$ and 0.568), 88.9% ($p = 0.930$ and $p > 0.999$), and 95.7% ($p = 0.244$ and 0.537)) (Fig. 2 and Supplementary Table 3). Further, the model had a comparative lesion-level sensitivity of 60.6% ($p = 0.107$ and 0.379), 64.3% ($p = 0.506$ and >0.999) and 59.6% ($p = 0.141$ and 0.329). In general, our model

had slightly higher patient-level sensitivity and NPV but without significant differences in the entire group, SAH and non-SAH subgroups, except for the patient-level sensitivity in the whole group for radiologists (significantly higher, $p = 0.037$), whereas the specificities were significantly lower than human clinicians ($p < 0.001$). Compared with the radiologists, the model showed superiority at a pre-specified 5% margin for patient-level sensitivity in the entire group and non-SAH subgroup, non-inferiority for patient-level sensitivity in the SAH subgroup and NPV in the entire group and non-SAH subgroup. Compared with the neurosurgeons, the model showed non-inferiority at a pre-specified 5% margin for patient-level sensitivity and NPV in the entire group and non-SAH subgroup. The mean diagnosis time per examination micro-averaged across radiologists and neurosurgeons were 30.1 s (95% CI: 29.2–31.0 s) and 22.4 s (95% CI: 21.1–23.6 s), respectively. While the model took 18.2 s (95% CI: 17.9–18.4 s) per case and was significantly faster than the radiologists ($p < 0.001$) but was comparable to the neurosurgeons ($p = 0.818$). Similar results were achieved in LYG data set, the external cohort. Compared to clinicians, the model demonstrated an improvement in patient-level sensitivity with superiority or non-inferiority at a pre-specified 5% margin. The micro-averaged diagnosis time was 27.1 s (95% CI: 26.3–28.0 s) and 25.7 s (95% CI: 24.1–27.3 s) for radiologists and neurosurgeons, respectively. The model took 19.6 s (95% CI: 19.3–20.0 s) per examination, which was significantly faster than radiologists ($p = 0.001$) and comparative to neurosurgeons ($p = 0.301$).

**Clinical application in the work-up of AIS in emergency department**. The results above demonstrate that our model had a higher lesion-level and patient-level sensitivity than the clinicians, which may have the potential for complementary implementation in clinical practice. Inspired by this finding, we wondered whether this model could work well when excluding the control cases to reduce workload in a specific clinical setting. The newly published guideline has strongly recommended CTA (with CT Perfusion) for AIS patients (Class I; level A) for selecting candidates for mechanical thrombectomy[15]. Radiologists are expected to detect stenotic and occlusive lesions and, as a result, they often overlook aneurysms. Intravenous thrombolysis is efficacious and safe for AIS patients, while it might increase risk of aneurysm rupture in some reports[32,33]. Therefore, radiologists should also pay special attention to the presence of IAs in this setting. So we endeavor to determine whether our model has the potential to apply to this population in excluding patients without IAs with high confidence, so that high-risk patients with IAs could be focused on more intensively. We enrolled another 333 consecutive patients from the emergency department of Jinling Hospital who had undergone emergent head CTA examination from 1 January 2019 to May 31, and from 1 September to 31 December 2019, and were suspicious for AIS (14 patients containing 16 IAs) (Internal cohort 5). The results demonstrate that our model had a specificity of 89.7%, moderate patient-level sensitivity of 78.6%, and NPV of 99.0%. From this result, we can assume that, with the triage of the model, 86.8% of patients were predicted as negative, among which 99.0% predicted-negative cases were true-negatives, and the other 13.2% were predicted as a high-risk group of having aneurysm(s). Therefore, radiologists can focus on these patients with more-intense attention in order to improve workflow and reduce workload (Fig. 3).

On the other hand, the 1.0% FN patients cannot be ignored. In this experiment, five aneurysms in three patients were missed. Three missed aneurysms were smaller than 4 mm and; three were located in MCA, and the other two were located in the ICA.

## Discussion

In this study, we developed and propose for clinical use a specific DL model to automatically predict and segment IAs in digital subtraction bone-removal CTA images and conducted a comprehensive translation study into the clinical scenarios to see the influence of occult cases, image quality, and manufacturers, which indicated the model's high tolerance. The validation process in the simulated real-world scenarios demonstrated that the model had higher patient-level sensitivity and lesion-level sensitivity than radiologists and neurosurgeons. Inspired by this, we further validated the model in the suspected AIS clinical scenario, which demonstrated that the model could exclude IA-negative cases with high confidence (99.0%) and help prioritize the clinical workflow to reduce the workload so that the radiologists could focus on high-risk patients. The study had a complete workflow of development and validation procedures from laboratory to real-world settings for clinical application. Implementation studies are warranted to develop appropriate and effective radiologists' alerts for the potentially critical finding of IAs and to assess their impact on reducing time to treatment.

There are several reasons contributing to the complexity of the IAs detection task. First, arterial visualization of CTA images is easily affected by the enhanced veins. Second, the prevalence of IAs in the general population and the SAH patients is significantly different (2–3% vs. 85%)[2,34]. Third, the IAs detection procedure includes two steps for clinicians, the first step is to identify cerebral arteries and then to recognize abnormal dilation (aneurysm), which is different from diagnosing solid lesions such as lung cancer[35], retinal diseases[18], and thyroid cancer[19]. The peculiarity of head CTA has resulted in little efforts that apply supervised learning to IA classification and detection. Because of these reasons, CAD of CTA to automatically detect IAs has been rarely reported, and only two studies exist in the literature[24,25]. The lack of a reference standard and external validation data or focusing only on non-ruptured aneurysms ≥3 mm limited the generalization and further application of their models in the clinical setting. In order to solve these issues, we enrolled a large number ($n = 1177$) of CTA cases for training the specific designed DL model for high robustness, which had been verified by DSA, the gold standard for IAs diagnosis, especially for small aneurysms. Besides, we collected external datasets from other hospitals for validation of generalizability. Other factors that influenced the model performance were also taken into consideration, including image quality, occult IAs, and different manufacturers. Our study also highlights the potential clinical application of the model in a suspected AIS setting, for which head CTA is recommended and the results demonstrated that the model could reduce radiologists' workload by triage. With all these efforts, our work provided a relatively complete and innovative insight into the development and validation of the model for automatic prediction and segmentation of IAs.

Several approaches were adopted to improve model performance. First, the proposed model was built based on the U-Net[30], a well-proven network structure that has been widely used in medical image segmentation. Second, we used the Basic Block instead of the stacked convolution layer, thus the performance of the deep conventional network was boosted by the residual connection[28]. Last, the dual attention block[29] was employed to enforce the network to focus on the informative region and features. We also compared the performance of the model to that of the most frequently employed 3D U-Net[36] using the same training and testing data (Internal cohort 1), and our model had a significantly higher performance (Supplementary Table 4).

Another strength of our proposed model is that all CTA studies in the training and, some of the internal and external validation cohorts were demonstrated by DSA results, which guaranteed the

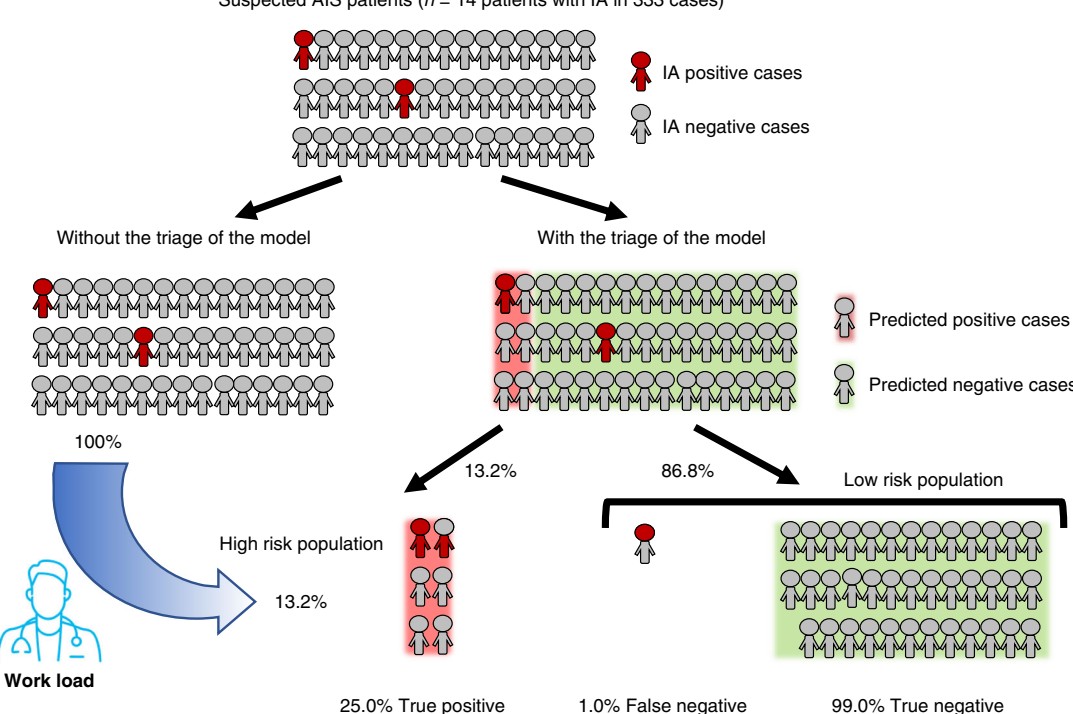

**Fig. 3 Impact of the proposed model on clinical practice for patients with suspicion of AIS in the emergency department.** In the cohort of patients with suspected AIS (Internal cohort 5), who were prescribed to perform head CTA examination, 86.8% patients diagnosed as aneurysm-negative cases by our model, among which 99.0% were true-negative, demonstrating high confidence in identifying negative cases by our model. As a result, only 13.2% of patients were categorized as high-risk, to whom the radiologists can pay more-intense attention and reduce their workload in detecting aneurysm in AIS patients. *AIS* acute ischemic stroke, *IA* intracranial aneurysm.

ground truth of the cases. Although inter-grader variability is a well-known issue in many settings including IAs diagnosis in CTA[11,12,37,38], human interpretation is still used as the reference standard in some studies[22–25]. Some structures, such as the infundibula, might otherwise be misdiagnosed as aneurysms. Unambiguous interpretations, which are called noisy labels can lead to an obviously biased performance of the model[39,40]. Therefore, it is necessary and urgent to use DSA, the gold standard for IAs detection in machine learning based CAD studies.

We also collected consecutive CTA images from other medical centers to test the generalizability of this AI tool. It is challenging for the model to achieve an excellent performance as the prevalence of patients with aneurysms in some cohorts was quite low and the CTA data in the validation sets were generated by multiple manufacturers that were different from the training set. Consequently, we have demonstrated that different manufacturers had an impact on the model's performances. Generally, our model has the potential to overcome some difficulties in the clinical practice of aneurysm detection, including a large number of diagnostic errors, substantial waste of resources, and inefficiencies in the workflow[41,42]. Given these model results, interpreters have the option to take the model results into consideration or disregard it based on their own judgment. With this approach to medical imaging, the model offers the visualized results that show the presence and location of the predicted IAs on CTA images at slice-level, in a fashion similar to clinical practice, which can be helpful to handle the results of the model when the results are different from radiologists. As a result, an interpretable representation is particularly useful in difficult and ambiguous cases. Such cases are commonly found in clinical practice and even specialized neuroradiology practitioners feel those difficult to detect.

Even though we have demonstrated high performance of the model, and the model can be run in real-time within or across entire hospital systems, we have to note the adverse effects on radiologists' workload and behaviors. For example, radiologists must respond to the FP predictions, which may result in increasing fatigue of radiologists and blunt human responsiveness to TP predictions by the model. In contrast, we have to avoid the unfavorable result of clinician overreliance on automated systems. On the other hand, there were still some visible aneurysms that were ignored by the model, such as those located in the ICA and MCA or those images generated by other manufacturers, and tiny aneurysms, to which radiologists must pay close attention when augmenting their interpretation with the tool.

The limitations of our study mainly lie in the relatively small sample of positive cases in the validation cohorts, which cannot be avoided because of the relatively lower prevalence of IAs in different clinical settings and hospitals. The second limitation is the data curation process, where CTA data with arteriovenous malformation/fistula (AVM/AVF) and head trauma were excluded. This means the algorithm may not perform as well for studies with subtle findings that a majority of radiologists would identify. Third, the performance of radiologists augmented by the model was not discussed in this study, as the process of device approval has not started yet. However, we can assume that the augmented human performance would be better than that without augmentation, which has been shown in one study by Park et al.[24]. Fourth, the model's patient-level sensitivity was not satisfying in some cohorts and different manufacturers can impact the performance of the model, so more-diverse training data may be required. Finally, we did not conduct a prospective multicenter controlled experiment to validate the model in clinical scenarios.

Notwithstanding these concerns, our proposed specific DL-based model for automated detection and segmentation of IAs had higher patient-level sensitivity and lesion-level sensitivity compared with radiologists and neurosurgeons, and it could reduce their workload. Several aspects of clinical effectiveness should be measured and tracked, including patient outcomes and costs. Therefore, it is a worthwhile venture to continue considerations aimed at optimal integration of the model within the routine radiology workflow in order to leverage the complementary strengths of the DL model with the clinician's Gestalt and experience.

## Methods

**Ethics and information governance.** This retrospective study was approved by the Institutional Review Board of Jinling Hospital, Medical School of Nanjing University, with a waiver of written informed consent. Only de-identified retrospective data were used for research, without the active involvement of patients.

**Data.** We performed a retrospective, multicohort, diagnostic study using raw cross-sectional digital subtraction head CTA image sets from four hospitals in China (Jinling Hospital, Tianjin First Central Hospital, Lianyungang First People's Hospital, and Nanjing Brain Hospital). As listed in Table 1 and Supplementary Table 1, eight cohorts were employed to devise and validate the model regardless of aneurysm rupture status.

Internal cohort 1 encompassed patients who underwent CTA examinations and were subsequently verified by cerebral DSA within 30 days in Jinling Hospital, Nanjing, China from 1 June 2009 to 31 March 2017. The data were then shuffled and separated into training/tuning/testing sets. The tuning set consisted of 100 cases (50 cases with aneurysms and 50 non-aneurysm controls), and the testing set had 150 cases with 50% cases having IAs. The tuning set was used to evaluate model performance at the end of each epoch during training and for hyper-parameter optimization, and the testing set was a held-out set of images used for evaluation of the trained models, never used by the algorithm during training or validation.

For internal validation sets, consecutive patients undergoing CTA examinations verified by DSA from 1 April 2017 to 31 December 2017 in Jinling Hospital (Internal cohort 2) were used for internal validation. Consecutive patients undergoing CTA examinations verified by DSA from 1 January 2018 to 31 May 2019 (Internal cohort 3) were collected for validation of the effect of image quality. Patients who underwent head CTA without DSA restraints from 1 June 2019 to 31 August 2019 in Jinling Hospital (Internal cohort 4) were included for simulated real-world validation and human-model comparison. Patients who were suspicious for AIS from 1 January 2019 to 31 May and from 1 September to 31 December 2019 (Internal cohort 5) were included for the function validation of whether the confident screening of aneurysm-negative cases can reduce radiologists' workload.

For external validation, DSA-verified consecutive eligible CTA cases from 1 January 2019 to 31 July 2019 in Nanjing Brain Hospital (NBH cohort) were enrolled for the effect validation of external data. Consecutive eligible CTA cases from Tianjin First Central Hospital were included for validation of the effect of different manufacturers in 2013–2018 (TJ cohort) to the model, which contained three different manufacturers including GE, Siemens and Toshiba. Consecutive eligible CTA cases from 1 August 2018 to 30 September 2019 in Lianyungang First People's Hospital (LYG cohort) were enrolled for simulated real-world validation and human-model comparison (Supplementary Table 1).

All CTA images sourced from four hospitals were in Digital Imaging and Communications in Medicine (DICOM) format. Five multidetector CT scanners (SOMATOM Definition, SOMATOM Definition Flash, and SOMATOM Definition AS+, GE Revolution CT, Toshiba Aquilion ONE) were used to generate source CTA images and all images were processed on a workstation (Syngo 2008G; Siemens) with the bone voxels removed by software (Neuro DSA application) in the core laboratory. The bone-removed DICOM images were used for annotation and training of models.

For patients who underwent CTA, verified by DSA, the interval was <30 days. A dedicated curation process was only conducted in the training data set (Internal cohort 1) for high-quality images; the exclusion criteria were as follows: (a) patients who underwent DSA before head CTA; (b) patients with >30-day interval between CTA and DSA; (c) patients who had surgical clips, coils, catheters, or other surgical hardware in the head; (d) patients with AVM/AVF, Moyamoya disease, arterial occlusive diseases, and other vasculopathies that affected the structure of the intracranial vasculature; (e) patients with incomplete image data, poor image quality, and unavailable images; (f) patients with IA on DSA but undetectable in CTA images. The control group included those who underwent CTA and DSA; all images were reviewed and were free of abnormal cerebral vasculopathies. Patients with infundibular dilations and vascular stricture lesions were also included as controls considering the negligible influence. Supplementary Fig. 2 shows the flowchart of this study.

In the validation cohorts with DSA verification, we only excluded patients who met exclusion criteria (a, b, c, e) and those with AVM/AVF. This latter vasculopathy was obvious to detect with a low prevalence and had a morphological influence on the intracranial arteries. This way we tested the applicability of the proposed model in a real-world clinical scenario. In those without DSA verification, we excluded patients who met exclusion criteria: (c, e), and those with AVM/AVF.

**Radiologist annotations.** The presence and locations of IAs in each patient were determined by DSA (the gold standard), if available. Specifically, the spatial resolution of cerebral CTA was inferior to that of DSA, therefore the locations of IAs on CTA were reviewed by three specialized neuroradiologists (Z.S., S.L., and C.S.Z. with 3, 8, and 13 years experiences in neuroradiology interpretation, respectively) with reference to the DSA images to establish the final ground truth. The observers were instructed to exclude CTA of poor quality and incomplete images that were not sufficient for diagnosis as well as CTA cases that were post-procedure.

For the patients without DSA verification, two specialized neuroradiologists (C.S.Z. and S.L.) had access to all the DICOM series, original reports, and clinical histories, as well as previous and follow-up examinations during interpretation to establish the best possible reference standard labels (the silver standard) and they were instructed to exclude CTA of poor quality. In the case of disagreement between the two observers, such as small aneurysms and infundibula, consensus was reached in a joint reading with the assistance of a senior neuroradiologist (L.J.Z. with 19 years of neuroimaging experience) and then the majority vote of three radiologists established reference standard labels.

After the explicit locations of IA for all examinations were determined, one neuroradiologist (Z.S.) annotated the IA on bone-removal CTA of DICOM series pixel-wise with Mimics software (Version 16.0). The neuroradiologist had access to all the DICOM series and the final standard to identify the accurate location of IAs on CTA. The identified aneurysms were manually segmented on sections that contained IAs on bone-removal CTA.

**Model development.** In this study, we developed a 3-dimensional (3D) CNN called DAResUNet for the segmentation of IAs from digital subtraction bone-removal CTA images to evaluate the presence and locations of aneurysms. DAResUNet is a CNN with an encoder–decoder architecture similar to 3D U-Net[30], which contains an encoding module (encoder) for abstracting contextual information and a symmetric decoding module (decoder) for expanding the encoded features to a full-resolution map with the same size and dimensionality as the input volume. We adopted residual blocks[28] to replace the original convolution blocks of U-Net to ensure stable training when the depth of the network was significantly increased. Besides, dilated convolutions were used in the top layer of the encoder to enlarge the receptive field of the network, as we only performed downsampling three times. To enhance the performance of the network by exploring long-range contextual information, we embedded a dual attention[29] module between the encoder and the decoder (Fig. 1a).

The size of the input volume of the DAResUNet was $80 \times 80 \times 80$, which was large enough to enclose the majority of IAs. 3D image patches with the above size were randomly sampled from the entire CTA volume during training. To balance the number of training samples containing and not containing aneurysms, sampled patches had a 50% probability to contain an aneurysm. Before patch sampling, data augmentations such as rotation, scaling, and flipping were applied to CTA scans.

Before reaching the network, inputs were clipped to [0, 900] Hounsfield units (Hu) and then normalized to [−1, 1]. The network was trained to optimize a weighted sum of a binary cross-entropy loss and a Dice loss. The Adam optimizer was used by setting the momentum and weight decay coefficients to 0.9 and 0.0001, respectively. We employed a poly learning rate policy, where the initial learning rate is multiplied by $\left(1 - \frac{iter}{total\ iter}\right)^{0.9}$ after each iteration. The initial learning rate was 0.0001 and the number of training epochs was 100[43]. In each epoch, we first selected 600 patients' images randomly from the training set, and then 100 patches including positive and negative samples were randomly cropped from each patient's image. In total, about 60,000 patches were used to train the model in each epoch.

In the inference stage (Fig. 1b), the segmentation prediction of the whole volume was generated by merging the prediction of uniformly sampled patches. Two adjacent patches may have 1/8 overlap, in other words, the strides along the three axes were all 40. For each voxel, we used the highest probability from all enclosing patches as its final prediction. In order to detect IAs in some low-contrast images clipped by the default window interval of [0, 900], another two intervals of [0, 450] and [−50, 650] were used to normalize the source images. The setting was automatically selected according to the brightness distribution. Given a bone-removal CTA image, a threshold value such as 150 Hu was used to find the initial area of vessels and then the maximum connectivity area was kept as the final region of vessels. Histogram of the brightness of voxels in the region was analyzed to find suitable clipping interval. We counted the distribution of three intervals including [0, 200], [200, 300], and [300, 500], which corresponded to clipping intervals of [0, 450], [−50, 650], and [0, 900], respectively. Finally, the clipping interval corresponding to the dominated distribution interval was selected to normalize the source images.

**Image quality dependence of the model performance**. Image quality was rated using multiplanar reconstructions, maximum intensity projections, and volume-rendered reformatted images[31]. Qualitative image scoring was performed independently by two neuroradiologists (Z.S. and S.L.) according to the degree of noise, vessel sharpness, and overall quality. In case of disagreement between both readers, consensus was reached in a joint reading to determine the final image quality score. CTA image quality was rated on a four-point scale.

**Human-model comparison experiment**. It is relatively easier to develop a DL model in a curated experimental environment than applying it in an ethical, legal, and morally responsible manner within a real-world healthcare setting. Therefore, we designed a validation process that simulated the real-world clinical procedures by applying the model in Internal cohort 4 and LYG cohort, which highlights the true context of the full breadth of CT scanning presented to clinicians from real-life clinical practice.

We performed a diagnostic accuracy study comparing performance metrics of radiologists with different years of experiences and the model. Six board-certified radiologists (two resident radiologists, F.X. and Y.X. with 4 years of working experience; two attending radiologists, L.W. and X.L.Z. with 7 years of working experience; and two assistant director radiologists, J.L. and Y.E.Z. with 11 and 13 years of working experience) and two neurosurgeons (L.L.W. with 10 years of working experience and L.J.Z. with 15 years of working experience) participating in the study were asked to interpret the CTA images and identify the presence of IAs. None of the six radiologists and two neurosurgeons were involved in the procedure of determining the location of IA as the reference standard. The clinicians read all the CTA images from Internal cohort 4 and LYG cohort. Acquired CTA image series were manually transferred to a dedicated workstation for review (Multi-Modality Workplace; Siemens Healthineer). CTA images were generated with a workstation (Syngo 2008G; Siemens) with the bone voxels removed by software (Neuro DSA application). The clinicians were blinded to clinical data as well as reference standard labels and independently analyzed all cerebral CTA images by using source images, maximum intensity projections, multiplanar reformations, volume-rendering reformatted images and target vessel reformation to determine the presence and locations of aneurysms, a way that simulates the real procedures of clinical practice. They were allowed to adjust, rotate, and reformat 3D images at the workstation to optimally view the presence and location of an individual aneurysm, while the model only used the bone-removal source images to generate IA predictions. They read independently in a diagnostic reading room, all using the same high-definition monitor (1920 × 1200 pixels) displaying CTA examinations on the dedicated workstation (Multi-Modality Workplace; Siemens Healthcare).

**Statistical analysis**. Quantitative variables were expressed as mean±SD if normally distributed, while median and inter-quartile range was used when non-normally distributed data. Categorical variables were expressed as frequencies or percentages. The segmentation model detects the potential aneurysm lesions in the CTA image data. To evaluate the performance of automatic segmentation, results were expressed as lesion-level sensitivity, Dice ratio, and false-positives per case (FPs/case); Dice ratio computed the overlap of the ground truth segmentation and the automatic segmentation. To evaluate the performance of automatic detection, accuracy, patient-level sensitivity, specificity, PPV, and NPV of the correct display of patients by the model were evaluated separately in each cohort and the 95% Wilson score confidence intervals were used to assess the variability. Bonferroni correction was applied to compare the metrics of the model among different image quality and manufacturers. The performance of clinicians was calculated in Internal cohort 4 and LYG cohort. The micro-average of patient-level sensitivity, specificity, accuracy, PPV, NPV, and lesion-level sensitivity across the radiologists and neurosurgeons were calculated by measuring each statistic pertaining to the total number of true-positive, true-negative, FP, and FN results. To assess model performance against those of the clinicians, we used a two-sided Pearson's chi-squared test or Fisher exact test to evaluate whether there were significant differences in specificity, patient-level sensitivity, accuracy, PPV, NPV, and lesion-level sensitivity. For comparisons with radiologists or neurosurgeons, the choice of superiority or non-inferiority was based on what seemed attainable from simulations conducted. The confidence limits of the difference were based on Wald method with Agresti-Caffo correction. For non-inferiority comparisons, a 5% absolute margin was pre-specified before the test set was inspected. Statistical analyses were conducted with SPSS Statistics (version 22.0.0, IBM SPSS Statistics, Armonk, New York), and R (version 3.5.2, R Foundation for Statistical Computing, Vienna, Austria). We used a statistical significance threshold of 0.05.

**Reporting summary**. Further information on research design is available in the Nature Research Reporting Summary linked to this article.

## Data availability
The data that support the findings of this study are available on request from the corresponding authors [L.J.Z. and G.M.L.]. The data with participant privacy/consent are not publicly available due to hospital regulation restrictions.

## Code availability
The code is available at https://github.com/deepwise-code/DLIA.

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

## Acknowledgements

This work was supported by the Key Projects of the National Natural Science Foundation of China (81830057 for L.J.Z.) and National Key Research and Development Program of China (2017YFC0113400 for L.J.Z.). We thank Drs. Fei Xia, Yuan Xie, Li Wang, Xiaolei Zhang, Jia Liu, and Yan'e Zhao for their effort in human interpretation of CTA examinations in Internal cohort 4 and LYG cohort; Xiang Kong, Wei Zhang, Li Qi, Weiwei Huang, Mengdi Li for DICOM data archive; Mengjie Lu for statistical analysis support and all those who had made contribution to the publication of the work.

## Author contributions

L.J.Z., G.M.L., X.L.L. initiated the project and the collaboration. C.W.P., H.W., X.L.L., Y.Z.Y. developed the network architectures, training, and testing setup. L.J.Z., Z.S., C.S.Z. designed the clinical setup. Z.S., C.S.Z., S.L. S.X., Y.G., Y.G.Z., C.C.M., X.C., B.H., W.D.X. created the data set and defined clinical labels. C.W.P., H.W. contributed to the software engineering. Z.S., C.S.Z., S.L. created the database. L.J.Z., X.Z., C.S.Z., S.L., Z.S., L.L.W., L.J.Z. contributed clinical expertize. Z.S., L.M., M.J.L., analyzed the data. L.J.Z., G.M.L., X.L.L. managed the project. Z.S., C.W.P., L.J.Z., S.U.J., S.R.H., D.D.M. wrote the paper. L.J.Z. contributed to the uncertainty estimation.

## Competing interests

Dr. Schoepf receives institutional research support from Bayer, Bracco, Guerbet, HeartFlow, Inc., and Siemens Healthineers and received personal fees for consulting and/or speaking from Bayer, Elucid BioImaging, General Electric, HeartFlow,. Inc., Keya Medical, and Siemens Healthineers. The other authors declare no competing interests.
