## [Peer Review File · Nature Communications]

Reviewers' Comments:

Reviewer #1:

Remarks to the Author:

Title:

- Title is appropriate.

Keywords:

- None provided.

Abstract:

- Lines 30 & 31: consider changing the following sentence "CTA is recommended as a standard diagnosis tool, while the interpretation is time-consuming and challenging" to something more readable like "CTA is recommended as the standard diagnosis tool; yet, interpretation can be time-consuming and challenging".

- Lines 35 & 36: consider changing the following sentence "Simulated real-world studies were conducted in consecutive internal and external cohorts, achieving improved sensitivity and negative predictive value than radiologists" to something more readable like "Simulated real-world studies were conducted in consecutive internal and external cohorts which achieved an improved sensitivity and negative predictive value compared to that of radiologists".

- Lines 36 - 39: consider changing the following sentence "A specific cohort of suspected acute ischemic stroke was employed and found 96.8% predicted-negative cases can be trusted with high confidence, leading to reducing in human burden" to something more readable like "A specific cohort of suspected acute ischemic stroke was employed and found 96.8% predicted-negative cases can be trusted with high confidence, leading to a potential reduction in human workload".

Introduction:

- Line 43: add "the" as follows: "with a prevalence of 3.2% in 'the' general population"

- Line 43 & 44: add "the" as follows: "in 'the' spontaneous subarachnoid hemorrhage (SAH) population"

- Line 48: change "diagnoses" to "diagnosis"

- Line 51: change "surgery" to "surgical"

- Line 54: change "tomographic" to "tomography"

- Line 55: please define AHA and ASA before using acronym

- Lines 58 & 59: change "imaging modality for the patients suspicious of SAH" to "imaging modality for patients suspicious for SAH" and change "in emergency department" to "in the emergency department".

- Lines 59 & 60: change "time-consuming and subspecialty-training-requiring" to "time-consuming and requires subspecialty training".

- Line 61: change the following sentence "variability and high false-positive (FP), false-negative (FN) rates" to "variability, high false-positive (FP), and false-negative (FN) rates".

- Line 62: change "aneurysms" to "aneurysm"

- Lines 62 - 66: change the following sentence "The diagnostic accuracy is dependent on several factors including aneurysms size, diversity of technological specifications (16- versus 64-detector rows), image acquisition protocols, image quality, image postprocessing algorithms and variations in radiologists' experiences, resulting in a mean sensitivity in the range of 70.7%–97.8% in detecting IAs" to "The diagnostic accuracy is dependent on several factors including aneurysm size, diversity of technological specifications (16- versus 64-detector rows), image acquisition protocols, image quality, image postprocessing algorithms and variations in radiologists' experiences. These factors result in a mean sensitivity in the range of 70.7%–97.8% in detecting IAs".

- Lines 73 - 74: change the following sentence "tools to help detect, increase efficiency and reduce disagreement among observers, finally potentially improving clinical care of the patients" to "tools to help detect, increase efficiency, and reduce disagreement among observers which may potentially improve clinical care of the patients"

- Lines 78 - 79: change the following sentence "thresholding, or a region-growing algorithm , while the performance and generalization are not satisfactory" to "thresholding, or a region-growing algorithm. Additionally, their performance and generalization are not satisfactory".
- Lines 79 - 81: change the following sentence "Nowadays, deep learning (DL) has shown significant potential in accurately detecting lesions on medical imaging and had reached or even superior to the expert-level of diagnosis" to "Nowadays, deep learning (DL) has shown significant potential in accurately detecting lesions on medical imaging and has reached, or perhaps surpassed, an expert-level of diagnosis"
- Lines 84 - 86: "While CTA based CAD system has been rarely reported, and only two recently published studies can be found, to the best of our knowledge" to "CTA based CAD systems have been rarely reported, our group was only able to find two recently published studies".
- Line 86: change "in" to "with a"
- Line 87 & 88: change "scenarios, thus they were not adequate to apply in real-world clinical settings, which may fall into the 'AI chasm'" to "scenarios; thus, they were not adequate to apply in real-world clinical settings and may fall into the 'AI chasm'"
- Line 97: change "detecting" to "detection"
- Line 102 & 103: change the following sentence "patients who had underwent cerebral digital subtraction angiography (DSA), the gold standard for diagnosing IAs, to verify the results of CTA in the training dataset" to "patients who underwent cerebral digital subtraction angiography (DSA) which is the gold standard for diagnosing IAs in order to verify the results of CTA in the training dataset"
- Line 104: please define CNN before using acronym

Results

- Lines 132, 133, & 137: change "tunning" to "tuning"
- Line 143: change "Totally" to "In all,"
- Line 143: If 4 aneurysms were missed in 4 patients but 2 patients had multiple IAs, how were only 4 aneurysms missed? Please address.
- Line 144 & 145: See above comment.
- Line 155: change "during Apr. 1, 2017 and Dec. 31, 2017" to "from Apr. 1, 2017 to Dec. 31, 2017"
- Line 156 & 157: change "The model reached accuracy, sensitivity, and specificity of 86.1%, 88.3%, and 84.3%, with a recall rate of 79.7% and FPs of 0.26/case" to "The model reached an accuracy, sensitivity, and specificity of 86.1%, 88.3%, and 84.3%, respectively, with a recall rate of 79.7% and FPs of 0.26/case"
- Line 158: change "Totally" to "In all,"
- Line 170: see above comment.
- Line 184 - 186: please reword the following sentence to make it readable: "These cases were abandoned in the Internal cohort 1 while remaining in the validation set in our original designation for that they were hard to annotate in CTA source images"
- Line 187: DL is not "magic power", please revise
- Line 219: please clarify the meaning of the phrase "For knowing the underlying causes of misclassified cases in our developed framework"
- Line 230 - 233: change "for that" to "as" and again reword the sentence to make more readable
- Line 252: change "had" to "have"
- Line 261: remove "the"
- Line 262: change "NVP" to "NPV"
- Line 296: "implication" or "implementation"?
- Line 297: change "wonder" to "wondered"
- Line 305: change "is" to "has"
- Line 306: add "on" after "focused"
- Line 312: add "for" after "suspicious"

Discussion:

- Line 360: change "the" to "these"

- Line 367: change "ones" to "aneurysms"
- Line 394: add "the" after "DSA,"
- Line 396: change "is" to "has"
- Lines 404 - 405: please revise the following sentence to make it readable "Such cases are common in clinics and even specialized neuroradiology practitioners are difficult to detect".
- Line 416: change "lay" to "lie"
- Line 422: change "for that" to "as"
- Line 424: remove "of"
- Lines 431 - 433: please revise the following sentence to make it more readable: "Therefore, further worthwhile consideration is how to best integrate the model with the routine radiology interpreting workflow, how to best leverage the complementary strengths of the DL framework and clinician gestalt and experience".

Methods:

- Lines 452 & 454: change "tunning" to "tuning"
- Line 466: change "Patients who suspected of AIS" to "Patients who were suspicious for AIS"

Tables & Figures:

- Tables and figures are appropriate.

Reviewer #2:

Remarks to the Author:

The work is convincing and the results relevant. Nothing very original from the computational or methodological point of view, although a very nice translation of DL techniques into the clinical scenarios. I think that the paper is worthy of publication, although further English-proofreading should be performed. Moreover, some comments are reported below:

Page 3: "The existing challenges also include inter-grader variability and high false-positive (FP), false-negative (FN) rates".

Report some figures and references, in order to weight the nature of the problem.

Page 4: This sentence is uncomplete: "While CTA based CAD system has been rarely reported, and only two recently published studies can be found, to the best of our knowledge^{25,26}".

Page 4: Although Topol's work is well known to me, the concept of "AI chasm" should be clarified to a larger audience of readers.

Abstract ant page 4: although intuitively clear, please rephrase and expand the meaning of "head bone-removal CTA images"

Page 4: "simulate the detecting procedure of the human brain". Re-phrase it. Moreover, is it CTA used to detect the "human brain"?

6: 2 were located in the cerebellar artery. In the cerebellar arteries as a whole or which one? Please specify.

Page 12: "which exposed the patients to the risk of aneurysm rupture if they have IAs"¹³ ?

"Clinical Application in Routine Practice and Comparison with Radiologists": I agree with the authors on the approach, but why not including neurosurgeons? The clinical application in routine practice often involves collaboration or even sole-management of neurosurgeons rather than radiologists. Without the involvement of neurosurgeons in this evaluation, the work can barely reach the real-world scenarios of aneurysms' detection and management.

In the discussion as well as in figure 5 it is not discussed the possibility that some of the missed small aneurysms might actually be infundibula. Some discussion on such a differential diagnosis should be included, as it's very clinically relevant.

"Model development": is there an empiric reason why the authors set the number of training epochs to 100?

Reviewer #3:

Remarks to the Author:

The manuscript describes the use of a 3D UNet-based deep learning framework for the segmentation of intracranial aneurysms in CTA images. The authors develop the method using a training set obtained in one hospital and perform an initial validation on images acquired in the same hospital. Several additional validation studies are performed in separately acquired data sets, with mixed results.

It's good that data was included from multiple hospitals, acquired with different CT scanners, and with different CT image quality. However, the manuscript also has some shortcomings. In general, the problem definition is unclear. Are these intracranial aneurysms before rupture or after rupture? Is the goal to perform segmentation or detection? Detection of small structures with deep learning is challenging, as is clear from the results in the paper for small lesions. The methodology used is not very novel and might also not be the optimal one here. If the goal is to detect these lesions, a regression-based localization approach (e.g. such as those used for landmark localization) might be better than segmentation. Overall, the sensitivity of the method is quite low and a method with this performance is unlikely to part of any clinical solution.

Major comments

- There is a high risk of bias in the dataset as the authors only include patients who have been imaged with both CTA and DSA. CTA imaging is much more common (see lines 124-125). Selecting only 11% of all CTA patients for the data set limits the applicability of the developed method as a screening tool in all CTA images.
- The class balance is quite different between the different datasets used. In the training set 869/1177 (74%) patients had IAs. In the tuning/test set, the prevalence of aneurysms is 50%. In the NBH cohort, 39/211 (18%) patients had IAs. In Internal cohort 2, the number IAs is not mentioned. In Internal cohort 3, 47/151 (31%) patients had IAs. Prevalence in Internal cohort 5 is much lower at 10/214 (~5%), as DSA was not a requirement in these patients. No probability calibration is performed to correct for this.
- In independent test sets (Internal cohort 2) sensitivity was substantially poorer than in the test set (88.3% vs. 97.3%) and even worse in the NBH cohort (82.1%) or in Internal cohort 5 (40%). This indicates poor generalizability of the method to new datasets.
- Throughout the paper, the authors list many values for sensitivity, specificity, recall, negative predictive value, accuracy, false positives/case. These are all quite connected and could be easily visualized in either a precision-recall plot, an ROC-curve, or an FROC-curve. For example, Fig.2 should be replaced with ROC curves that allow the reader to assess the differences between the automatic algorithm and the human observers. E.g., in Lines 294-296 the authors emphasize how their method has better sensitivity than humans. However, Fig. 2 shows that specificity and accuracy are lower than for humans. By picking a different operating point on the ROC curve, you can always find some criterion (sensitivity or specificity) on which you outperform the human, but what does that really mean?
- In Lines 312-319 and Fig. 3 the authors describe how in Internal cohort 5, the method identified 12.6% of patients as containing an aneurysm, and how this is a good result as the radiologists do not have to focus on the other 87.4% of patients. However, in this cohort, only 5% of patients had aneurysms anyway, and the sensitivity is only 40%. Hence, 60% of aneurysms were missed by

the algorithm and thus by a radiologist relying on this algorithm. This is a very poor result in this consecutive cohort.

- Line 170: In the NBH cohort, the method misses IAs in one in three patients.
- Lines 237-249. Were these patients from the 'general population', for whom DSA images were not always available?
- It would be good to include more images of cases in which the method worked or did not work well.
- Lines 130-131: It's unclear what was segmented in the images.

Minor comments

- Line 127-128: Why do the authors mention the total number of 2D slices? This is irrelevant as the method operates on 3D image patches.
- Line 140: Number of FPs is set to be 0.3, but what does that mean? 0.3 FP per CTA scan?
- Lines 156-157: What do the authors consider to be the difference between sensitivity and recall rate? Aren't these the same?
- Lines 177-178: Why are there three p values for two cohorts? What kind of statistical test was used here?
- Lines 200-203: I don't really see a pattern between image quality and model performance. Was there any?
- Lines 206-207. Barely is a bit exaggeration. 10/151 is still around 6%.
- Line 250: What is meant with microaverage sensitivity?
- Typos, grammar, e.g.
 - o Line 145. Comma after 'That is'
 - o Line 211: patents > patients
 - o Line 258: NVP > NPV
 - o Line 293: AIS > IAS
 - o Line 452: running > tuning

Department of Diagnostic Radiology
Jinling Hospital, Medical School of Nanjing University
305 Zhongshan East Road, Nanjing
Jiangsu Province, 210002, China
Email: kevinzhlj@163.com

September 1, 2020

Dear referees,

We would like to give a lot of thanks to the referees for your helpful and insightful comments for our manuscript (**NCOMMS-20-10696-T**) entitled “Clinically Applicable Deep Learning for Intracranial Aneurysm Detection in Computed Tomography Angiography Images: A Comprehensive Multicohort Study”, which have significantly improved the manuscript. We have taken all comments seriously and carefully revised the manuscript according to the suggestions.

Our detailed responses to the specific comments are presented in the next pages. The original comments are in *a red italic font*, and our responses are in a black regular font.

Again, thank you very much and look forward to hearing from you soon!

Sincerely yours,

Long Jiang Zhang, M.D., Ph.D.,

Reviewer #1 (Remarks to the Author):

R1-1. Title:

- Title is appropriate.

Keywords:

- None provided.

Abstract:

- Lines 30 & 31: consider changing the following sentence “CTA is recommended as a standard diagnosis tool, while the interpretation is time-consuming and challenging” to something more readable like “CTA is recommended as the standard diagnosis tool; yet, interpretation can be time-consuming and challenging”.
- Lines 35 & 36: consider changing the following sentence “Simulated real-world studies were conducted in consecutive internal and external cohorts, achieving improved sensitivity and negative predictive value than radiologists” to something more readable like “Simulated real-world studies were conducted in consecutive internal and external cohorts which achieved an improved sensitivity and negative predictive value compared to that of radiologists”.
- Lines 36 - 39: consider changing the following sentence “A specific cohort of suspected acute ischemic stroke was employed and found 96.8% predicted-negative cases can be trusted with high confidence, leading to reducing in human burden” to something more readable like “A specific cohort of suspected acute ischemic stroke was employed and found 96.8% predicted-negative cases can be trusted with high confidence, leading to a potential reduction in human workload”.

Introduction:

- Line 43: add “the” as follows: “with a prevalence of 3.2% in ‘the’ general population”
- Line 43 & 44: add “the” as follows: “in ‘the’ spontaneous subarachnoid hemorrhage (SAH) population”
- Line 48: change “diagnoses” to “diagnosis”
- Line 51: change “surgery” to “surgical”
- Line 54: change “tomographic” to tomography”
- Line 55: please define AHA and ASA before using acronym
- Lines 58 & 59: change “imaging modality for the patients suspicious of SAH” to “imaging modality for patients suspicious for SAH” and change “in emergency department” to “in the emergency department”.
- Lines 59 & 60: change “time-consuming and subspecialty-training-requiring” to “time-consuming and requires subspecialty training”.
- Line 61: change the following sentence “variability and high false-positive (FP), false-negative (FN) rates” to “variability, high false-positive (FP), and false-negative (FN) rates”.
- Line 62: change “aneurysms” to “aneurysm”
- Lines 62 - 66: change the following sentence “The diagnostic accuracy is dependent on several factors including aneurysms size, diversity of technological specifications (16-

versus 64-detector rows), image acquisition protocols, image quality, image postprocessing algorithms and variations in radiologists' experiences, resulting in a mean sensitivity in the range of 70.7%–97.8% in detecting IAs” to “The diagnostic accuracy is dependent on several factors including aneurysm size, diversity of technological specifications (16- versus 64-detector rows), image acquisition protocols, image quality, image postprocessing algorithms and variations in radiologists' experiences. These factors result in a mean sensitivity in the range of 70.7%–97.8% in detecting IAs ” .

- Lines 73 - 74: change the following sentence “tools to help detect, increase efficiency and reduce disagreement among observers, finally potentially improving clinical care of the patients” to “tools to help detect, increase efficiency, and reduce disagreement among observers which may potentially improve clinical care of the patients”

- Lines 78 - 79: change the following sentence “thresholding, or a region-growing algorithm , while the performance and generalization are not satisfactory” to “thresholding, or a region-growing algorithm. Additionally, their performance and generalization are not satisfactory”.

- Lines 79 - 81: change the following sentence “Nowadays, deep learning (DL) has shown significant potential in accurately detecting lesions on medical imaging and had reached or even superior to the expert-level of diagnosis” to “Nowadays, deep learning (DL) has shown significant potential in accurately detecting lesions on medical imaging and has reached, or perhaps surpassed, an expert-level of diagnosis”

- Lines 84 - 86: “While CTA based CAD system has been rarely reported, and only two recently published studies can be found, to the best of our knowledge” to “CTA based CAD systems have been rarely reported, our group was only able to find two recently published studies”.

- Line 86: change “in” to “with a”

- Line 87 & 88: change “scenarios, thus they were not adequate to apply in real-world clinical settings, which may fall into the ‘AI chasm”” to “scenarios; thus, they were not adequate to apply in real-world clinical settings and may fall into the ‘AI chasm””

- Line 97: change “detecting” to “detection”

- Line 102 & 103: change the following sentence “patients who had underwent cerebral digital subtraction angiography (DSA), the gold standard for diagnosing IAs, to verify the results of CTA in the training dataset” to “patients who underwent cerebral digital subtraction angiography (DSA) which is the gold standard for diagnosing IAs in order to verify the results of CTA in the training dataset”

- Line 104: please define CNN before using acronym

Results

- Lines 132, 133, & 137: change “tunning” to “tuning”

- Line 143: change “Totally” to “In all,”

Response:

Highly appreciate your comments. We have revised the manuscript as suggested and added Keywords following the part of Abstract (see the red highlight in the revision):

“Key words: Intracranial aneurysms; Computed tomographic angiography; Deep learning; Object detection; Medical image segmentation; Real word”

R1-2. - Line 143: *If 4 aneurysms were missed in 4 patients but 2 patients had multiple IAs, how were only 4 aneurysms missed? Please address*

Response:

We clarified this issue in this revised manuscript. For the 4 patients with missed aneurysms, two had solitary aneurysm, one had 2 aneurysms, and one had 3 aneurysms. The model missed one aneurysm in the 4 patients, respectively, see the following **Table R1**.

Table R1. Overview of the missed aneurysms in the testing set.

	Case 1	Case 2	Case 3	Case 4
No. of aneurysms for ground truth	1	1	2	3
No. of aneurysms the model detected	0	0	1	2
No. of aneurysms the model missed	1	1	1	1
Patient-based result	False	False	True	True
Aneurysm-based result	False	False	False	False

R1-3. - Line 144 & 145: *See above comment*

Response:

Our response can be seen in our response to R1-2 and the revised manuscript.

R1-4. - Line 155: *change “during Apr. 1, 2017 and Dec. 31, 2017” to “from Apr. 1, 2017 to Dec. 31, 2017”*

- Line 156 & 157: *change “The model reached accuracy, sensitivity, and specificity of 86.1%, 88.3%, and 84.3%, with a recall rate of 79.7% and FPs of 0.26/case” to “The model reached an accuracy, sensitivity, and specificity of 86.1%, 88.3%, and 84.3%, respectively, with a recall rate of 79.7% and FPs of 0.26/case”*

- Line 158: *change “Totally” to “In all,”*

- Line 170: *see above comment.*

Response:

Revised as suggested.

R1-5. - Line 184 - 186: *please reword the following sentence to make it readable: “These cases were abandoned in the Internal cohort 1 while remaining in the validation set in our original designation for that they were hard to annotate in CTA source images”*

Response:

Thanks for your suggestion. We have reworded the sentence to “These cases were excluded in the internal cohort 1 and the validation cohorts because they were impossible to be annotated in CTA source images”. One example was presented in **Fig. R1** as below.

Fig. R1. An example of the occult case which cannot be detected in CTA. This patient had an aneurysm in the ophthalmic segment of the left internal carotid artery. **a.** Volume-rendered CT angiography image. **b.** Volume-rendered 3D DSA image. The yellow circle (right) denotes an aneurysm in 3D-DSA image, while CTA image does not show the aneurysm (white circle), Thus, the aneurysm cannot be annotated in source CTA images.

R1-6. - Line 187: DL is not “magic power”, please revise.

Response:

Revised as suggested.

R1-7. - Line 219: please clarify the meaning of the phrase “For knowing the underlying causes of misclassified cases in our developed framework”

Response:

Thanks for your suggestion. We have revised it as “To uncover the underlying causes of misclassified cases in our developed framework”.

R1-8. - Line 230 - 233: change “for that” to “as” and again reword the sentence to make more readable

Response:

Thanks for your suggestions. We have revised the manuscript as you suggested and reworded the sentence as the following (see the red highlight in the revision):

“We defined unexplained reasons as these aneurysms that were obvious for radiologists to identify but were missed by our framework. The unexplained cases were mainly found in the TJ cohort and NBH cohort, which may be attributed to different manufacturers or CTA scan protocols.”

R1-9. - Line 252: change “had” to “have”

- Line 261: remove “the”

- Line 262: change “NVP” to “NPV”

- Line 296: “implication” or “implementation”?

- Line 297: change "wonder" to "wondered"
- Line 305: change "is" to "has"
- Line 306: add "on" after "focused"
- Line 312: add "for" after "suspicious"

Discussion:

- Line 360: change "the" to "these"
- Line 367: change "ones" to "aneurysms"
- Line 394: add "the" after "DSA,"
- Line 396: change "is" to "has"

Response:

Thanks for your suggestions. Revised as suggested.

R1-10. - Lines 404 - 405: please revise the following sentence to make it readable "Such cases are common in clinics and even specialized neuroradiology practitioners are difficult to detect".

Response:

Thanks for your suggestions. We have revised the manuscript as you suggested and reworded the sentence as the following (see the red highlight in the revision):

"Such cases are commonly found in clinical practice and even specialized neuroradiology practitioners feel those difficult to detect."

- R1-11. - Line 416: change "lay" to "lie"
- Line 422: change "for that" to "as"
- Line 424: remove "of"

Response:

Revised as suggested.

R1-12. - Lines 431 - 433: please revise the following sentence to make it more readable: "Therefore, further worthwhile consideration is how to best integrate the model with the routine radiology interpreting workflow, how to best leverage the complementary strengths of the DL framework and clinician gestalt and experience".

Response:

Thanks for your suggestions. We have revised the manuscript as you suggested and reworded the sentence as the following (see the red highlight in the revision):

"Therefore, it is a worthwhile venture to continue considerations aimed at optimal integration of the model within the routine radiology workflow in order to leverage the complementary strengths of the DL framework with the clinician's Gestalt and experience".

R1-13. *Methods:*

- Lines 452 & 454: change "tunning" to "tuning"
- Line 466: change "Patients who suspected of AIS" to "Patients who were suspicious for AIS"

Tables & Figures:

- Tables and figures are appropriate.

Response:

Revised as suggested.

Reviewer #2 (Remarks to the Author):

The work is convincing and the results relevant. Nothing very original from the computational or methodological point of view, although a very nice translation of DL techniques into the clinical scenarios. I think that the paper is worthy of publication, although further English-proofreading should be performed.

Response:

Thank you for your comments and encouragement. We have carefully revised the manuscript. Prof. U. Joseph Schoepf, Mr. Rock H. Savage, and Mr. Danielle M. Dargis, three native English speakers from Medical University of South Carolina, help us polish the full text. We hope you will find our revised manuscript satisfactory.

R2-1. Page 3: "The existing challenges also include inter-grader variability and high false-positive (FP), false-negative (FN) rates".

Report some figures and references, in order to weight the nature of the problem.

Response:

We reworded the sentence as the following (see the red highlight in the revision): "The existing challenges also include inter-observer variability and high false-negative (FN) rates". We also added the following 3 references.

References

1. Lubicz, B., et al. Sixty-four-row multisection CT angiography for detection and evaluation of ruptured intracranial aneurysms: Interobserver and intertechnique reproducibility. *AJNR Am J Neuroradiol* 28,1949-1955 (2007).
2. Maldaner, N., et al. Interrater agreement in the radiologic characterization of ruptured intracranial aneurysms based on computed tomography angiography. *World Neurosurg* 103,876-882 (2017).
3. Bechan, R.S., et al. CT angiography versus 3D rotational angiography in patients with subarachnoid hemorrhage. *Neuroradiology* 57,1239–1246 (2015).

R2-2. Page 4: This sentence is uncomplete: "While CTA based CAD system has been rarely reported, and only two recently published studies can be found, to the best of our knowledge 25,26".

Response:

Thank you for your comments. We reworded it as the following (see the red highlight in the revision): "CTA based CAD systems for automatically detecting intracranial aneurysms have been rarely reported, and only two recently published studies can be found in this field, to the best of our knowledge 25,26".

R2-3. Page 4: Although Topol's work is well known to me, the concept of "AI chasm" should be clarified to a larger audience of readers.

Response:

Thank you for your comments. We have added the concept of "AI chasm" in the revised manuscript as following: "(AI chasm,) which can be described as a divide between developing a scientifically sound algorithm and its use in any meaningful real-world application¹."

References:

1. United States Food and Drug Administration (FDA) Medical Devices. Evaluation of Automatic Class III Designation (De Novo). <https://www.fda.gov/MedicalDevices/DeviceRegulationandGuidance/HowtoMarketYourDevice/PremarketSubmissions/ucm462775.htm> (2018).

R2-4. Abstract ant page 4: although intuitively clear, please rephrase and expand the meaning of "head bone-removal CTA images"

Response:

Thank you for your comments. We have revised "head bone-removal CTA images" to "1,177 digital subtraction head bone-removal CTA images, which were based on a section-by-section subtraction to subtract nonenhanced from enhanced CT data to facilitate the diagnosis of aneurysms."¹

Reference

1. Tomandl, B.F., et al. Bone-subtraction CT angiography for the evaluation of intracranial aneurysms. *AJNR Am J Neuroradiol* 27,55-59 (2006).

R2-5. Page 4: "simulate the detecting procedure of the human brain". Re-phrase it. Moreover, is it CTA used to detect the "human brain"?

Response:

Thank you for your comments. We have revised the sentence as "Thus, we submit it can simulate the detection procedure of the human brain".

R2-6. 6: 2 were located in the cerebellar artery. In the cerebellar arteries as a whole or which one? Please specify.

Response:

Thanks for your comments. The cerebellar arteries include the superior cerebellar artery, the anterior inferior cerebellar artery, and the posterior inferior cerebellar artery. The prevalence of aneurysm in the cerebellar arteries is extremely low¹, so we regarded the cerebellar arteries as a whole for analysis. In this study, the 2 missed aneurysms in cerebellar artery were located in the right posterior inferior cerebellar artery and the left superior cerebellar artery.

Reference

1. Imaizumi, Y., Mizutani, T., Shimizu, K., Sato, Y., Taguchi, J. Detection rates and sites of unruptured intracranial aneurysms according to sex and age: an analysis of MR angiography-based brain examinations of 4070 healthy Japanese adults. *J Neurosurg* 130,573-578 (2018).

R2-7. Page 12: “which exposed the patients to the risk of aneurysm rupture if they have IAs”¹³ ?

Response:

Thanks for your suggestions. We have reworded the sentence as the following (see the red highlight in the revision): Administration of antiplatelet or anticoagulant therapy is often recommended, which will increase risk of aneurysm rupture¹³.

R2-8. “Clinical Application in Routine Practice and Comparison with Radiologists”: I agree with the authors on the approach, but why not including neurosurgeons? The clinical application in routine practice often involves collaboration or even sole-management of neurosurgeons rather than radiologists. Without the involvement of neurosurgeons in this evaluation, the work can barely reach the real-world scenarios of aneurysms’ detection and management.

Response:

We highly appreciate your suggestions. As suggested, we invited 2 neurosurgeons to participate this study in the revised manuscript.

R2-9. In the discussion as well as in figure 5 it is not discussed the possibility that some of the missed small aneurysms might actually be infundibula. Some discussion on such a differential diagnosis should be included, as it’s very clinically relevant.

Response:

Thanks for your excellent comments. We reviewed all missed aneurysms and found that all missed aneurysms were not infundibula. Infundibulum is a focal, symmetric, conical dilatation at the origin of a blood vessel that can easily be mistaken for a small aneurysm¹. An infundibulum is small, typically less than 3 mm in diameter. The distal vessel typically arises from the apex—not the side—of the infundibulum. So, a typical infundibulum is relatively easy to distinguish from an aneurysm especially for DSA. In this study, in the cohorts without DSA verification, most possible reference standard labels (the silver standard) were established by two specialized neuroradiologists. And in the case of disagreement between the two observers, consensus was reached in a joint reading with the assist of a senior neuroradiologist and then the majority vote of 3 radiologists established reference standard labels. So some of the missed small aneurysms might actually be infundibula is scarce. However, we discussed this issue you mentioned in the Discussion section in this revised manuscript.

Reference

1. Anne G Osborn. Brain: Imaging, pathology, and anatomy. Second Edition

R2-10. “Model development”: is there an empiric reason why the authors set the number of training epochs to 100?

Response:

Highly appreciate your excellent comment. It is just an experience value and the number of training epochs to 100 is frequently used by other researchers^{1,2}. During experiments, we trained our model from scratch using different settings, and we found that all training

procedures could reach convergence within 100 epochs. In each epoch, we first randomly selected 600 patients' images from training set, and then 100 patches including positive and negative samples were randomly cropped from each patient's images. In total, about 60,000 patches were used to train the model in each epoch.

References

1. Lan, Y., Xiang, Y., Zhang L.C. An elastic interaction-based loss function for medical image segmentation. arXiv 02663 (2007).
2. Mehta, S., et al. Y-Net: Joint segmentation and classification for diagnosis of breast biopsy images. arXiv 1806.01313 (2018).

Reviewer #3 (Remarks to the Author):

R3-1. The manuscript describes the use of a 3D UNet-based deep learning framework for the segmentation of intracranial aneurysms in CTA images. The authors develop the method using a training set obtained in one hospital and perform an initial validation on images acquired in the same hospital. Several additional validation studies are performed in separately acquired data sets, with mixed results. It's good that data was included from multiple hospitals, acquired with different CT scanners, and with different CT image quality.

Response:

Thank you for your comments.

R3-2. However, the manuscript also has some shortcomings. In general, the problem definition is unclear. Are these intracranial aneurysms before rupture or after rupture? Is the goal to perform segmentation or detection?

Response:

Thanks for your comments. The study aimed to develop a robust and reliable AI tool for CAD of IAs in a clinical real-world application, regardless of ruptured and unruptured status. The purpose of our study was detection of intracranial aneurysms. Specifically, given a head or head/neck CTA scan, our goal is to evaluate the presence and location of intracranial aneurysms. In our study, segmentation-based methods were used to achieve the purpose. By analyzing the segmentation results, we can evaluate the presence, numbers and location of intracranial aneurysm segmented.

R3-3. Detection of small structures with deep learning is challenging, as is clear from the results in the paper for small lesions. The methodology used is not very novel and might also not be the optimal one here. If the goal is to detect these lesions, a regression-based localization approach (e.g. such as those used for landmark localization) might be better than segmentation. Overall, the sensitivity of the method is quite low and a method with this performance is unlikely to part of any clinical solution.

Response:

Thank you for your excellent comments. We have carefully revised the manuscript as suggested. We frankly acknowledged that detection and segmentation of small IAs are challenging. In this study, we performed both segmentation and detection tasks for further

real-world application and facilitate the clinical usage. We carefully double checked the misdiagnosed cases and found that some CTA images had inappropriate window width of 900 Hu and window level of 450 Hu [450,900], which could be the cause of the misdiagnosis (**Fig. R2**).

Fig. R2. Examples of the misdiagnosed cases with inappropriate window width and window level. **Panel a**, a patient with two aneurysms in the apex of vertebral basilar artery (arrow). **Panel b**, a patient with an aneurysm in anterior communication artery (arrow). Both CT images were set at the window width of 900 Hu and window level of 450 Hu [450,900]. After adjusting the window width and window level to [350,700] and [225,450], the images become brighter and the aneurysms are more obvious, and the aneurysms can be recalled by the model.

Therefore, in order to detect IAs in some low-contrast images clipped by the default window width and window level, another two intervals containing [0, 450] and [-50, 650] were used to normalize the source images. The setting was automatically selected according to the brightness distribution (please see **Model development** part in the **Online methods**). The results are listed in the **Table R2**, which is encouraging.

Table R2. Performance of the classifier in the cohorts.

	Initial Performance		Increased correct predictions		Updated Performance	
	Sensitivity	recall	Patient(s)	Aneurysm(s)	Sensitivity	recall
Internal cohort 1	97.3%	95.6%	0	0	97.3%	95.6%
Internal cohort 2	88.3%	79.7%	2	2	94.4%	84.1
Internal cohort 3	82.6%	74.6%	2	3	87.0%	79.7%
Internal cohort 4	69.8%	59.2%	2	2	73.6%	60.6%
Internal cohort 5	40.0%	33.3%	3	3	70.0%	58.3%
TJ cohort	64.1%	64.0%	3	3	71.8%	70.0%

NBH cohort	82.1%	73.9%	1	1	84.6%	76.1%
LYG cohort	81.7%	74.7%	1	2	85.0%	78.9%

Besides, we also noticed that the sample size in some cohorts is relatively small. So we have enrolled the consecutive cases in additional temporally or spatially independent data sets for comprehensive validation. The added data are shown in **Table R3**.

Table R3. The overview of the added datasets used for training, validation and testing of the framework

Number	Cohort	Patients, n	Added patients	Total	Configuration, n (%)	
					Cases with IA (%)	Control (%)
#1	Internal cohort 1	1177	0	1177	869 (73.8)	308 (26.2)
#2	Internal cohort 2	245	0	245	108 (44.1)	137 (55.9)
#3	Internal cohort 3	151	75	226	61 (27.0)	165 (73.0)
#4	Internal cohort 4	374	0	374	53 (14.2)	321 (85.8)
#5	Internal cohort 5	214	119	333	14 (4.2)	319 (95.8)
#6	NBH cohort	211	0	211	39 (18.5)	172 (81.5)
#7	TJ cohort	59	88	147	109 (74.1)	38 (25.9)
#8	LYG cohort	316	0	316	60 (19.0)	256 (81.0)
Total		2748	282	3029	1313 (43.3)	1716 (56.7)

We would like to further clarify the novelty in methodological point and experiment design for this study.

From the computational or methodological point of view, our model was well designed. First, an encoder-decoder architecture like U-Net¹ was adopted for smooth and gradual transitions from medical images to segmentation mask. Second, residual blocks instead of the stacked convolutions were used to make the stable training for increasing depth of the network. Third, a non-local attention block was embedded in 3D space to capture more reliable feature representations with long-range contextual information. We also compared the performances of the framework to that of the most frequently employed 3D U-Net² using the same training and testing data (Internal cohort 1), and our framework had significantly higher performance (see **Extended Data Table 4** in the manuscript). It also can be found in **Table R4** below to illustrate the superiority of our model (DAResU-Net).

In our study, we used segmentation-based methods to detect IAs due to the following reasons. First, we had complete aneurysm mask annotations, in other word, we knew which voxels in the CTA volume belong to aneurysms. Second, segmentation methods can take full advantage of the voxel-level annotation labels. Finally, for clinical use, segmented mask had more applications like shape analysis and size measurement. For

regression-based localization approach especially landmark localization, the number of landmarks is usually fixed and known in advance, which is not consistent with our task.

On the other hand, we designed a relatively integrated workflow of the validation process to highlight the translation of DL techniques into the clinical scenarios (see **Fig. R3**). Firstly, internal and external validation cohorts were applied to demonstrate the performance of the model. Then we conducted a comprehensive analysis of the influence of occult cases, image quality and manufacturers, which indicated the framework's relatively high tolerance and the potential of clinical application. Next, the validation process in the simulated real-world scenarios demonstrated that the framework had higher sensitivity and NPV than radiologists. We further validated the framework in the suspected acute ischemia stroke setting, in which head CTA is recommended by AHA/ASA guideline³. Our study demonstrated that the framework could exclude IA-negative cases with high confidence. All the procedures demonstrated the integrity of the study design.

Lastly, there are some challenges we have to face: firstly, the prevalence of patients with aneurysms was quite lower in Internal cohort 3 (prevalence of 27.0%), 4 (14.2%) and 5 (4.2%), as well as LYG cohort (19.0%), which means the performances of the model can be easily affected by the number of misdiagnosed aneurysms. Secondly, the CTA data in the training set were generated by Siemens Somatom Definition Flash or Somatom Definition with slice thickness of 0.5 mm. While the validation datasets from other hospitals are with different manufacturers, such as the CTA images in the NBH cohort were generated by GE Optima 660 and SIEMENS SOMATOM Definition AS+ and the slice thickness is 0.625 mm and 0.6 mm, respectively. TJ cohort had three different manufacturers (SIEMENS, GE and Toshiba). Our results demonstrated that different manufacturers have a great impact on the model performances, in which the model had a high performance in SIEMENS-generated CTA while lower performance was found in GE or Toshiba. Thirdly, in cohort 5, we aimed to study the framework can work well when excluding the control cases to reduce workload in AIS setting, for which the prevalence of aneurysms is very low (4.2%), thus the model may have the potential for complementary implementation in clinical practice. In the revised manuscript, we have added cases and the results are encouraging, which indicates the potential for clinical application in real-world environment.

References

1. Çiçek, Ö., Abdulkadir, A., Lienkamp, S.S., Brox, T., Ronneberger, O. 3D U-Net: learning dense volumetric segmentation from sparse annotation. International Conference on Medical Image Computing and Computer-Assisted Intervention, 424–432.
2. Adrian, W. Wolny/pytorch-3dunet: PyTorch implementation of 3D U-Net (Version v1.0.0). Preprint at <http://doi.org/10.5281/zenodo.2671581> (2019).
3. Powers, W.J. et al. Guidelines for the early management of patients with acute ischemic stroke: 2019 Update to the 2018 guidelines for the early management of acute ischemic stroke: A guideline for healthcare professionals from the American Heart Association/American Stroke Association. *Stroke* 50, e344-e418 (2019).

Table R4. Comparison of the performance of the framework to that of the most frequently employed 3D U-net model using the same training data (Internal cohort 1)

Model	Accuracy	Sensitivity	Specificity	PPV	NPV	Recall	Dice
U-Net_3D	73.3% (65.7%-79.8%)	94.7% (87.1%-97.9%)	52.0% (40.9%-62.9%)	66.4% (57.0%-74.6%)	90.7% (78.4%-96.3%)	92.2% (84.8%-96.2%)	0.666 (0.611-0.721)
DAResU-Net	86.0% (79.5%-90.7%)	97.3% (90.8%-99.3%)	74.7% (63.8%-83.1%)	79.4% (70.0%-86.4%)	96.6% (88.3%-99.0%)	95.6% (89.1%-98.3%)	0.752 (0.708-0.796)
p	0.006	0.405	0.004	0.041	0.421	0.351	0.006

The data in parentheses are 95% confidence interval;

NPV, negative predictive value; PPV, positive predictive value. The validated results at $P < 0.05$ are in bold and italic.

Fig. R3. Workflow of the design of the study and the prospect.

R3-4. There is a high risk of bias in the dataset as the authors only include patients who have been imaged with both CTA and DSA. CTA imaging is much more common (see lines 124-125). Selecting only 11% of all CTA patients for the data set limits the applicability of the developed method as a screening tool in all CTA images.

Response:

Thanks for your excellent comments.

Aneurysm detection in CTA images is challenging, especially for the small IAs. For patients without DSA verification, the ground truth would be identified by the cooperation of a team of experts based on CTA imaging. While another fact is that “CTA interpretation is time-consuming and requires subspecialty training. The existing challenges also include inter-observer variability and high false-negative (FN) rates”,¹⁻³ which would result in some unambiguous or wrong interpretations. And it is well known that unambiguous or wrong interpretations can lead to an obviously biased performance of the model.⁴⁻⁵ Therefore, we regarded as DSA should be gold standard for detecting intracranial aneurysms. Importantly, the sample size in training cohort in our study is 1177 cases with DSA, which is the largest sample size, to the best of our knowledge. Based on the above-mentioned considerations, CTA images with DSA validation to train the deep learning models should be the preferred choice.

On the other hand, we also included patients without DSA in the testing procedures (part of *Clinical Application in Routine Practice and Comparison with Radiologists*) in the real-world practice.

References

1. Lubicz, B., et al. Sixty-four-row multisection CT angiography for detection and evaluation of ruptured intracranial aneurysms: interobserver and intertechnique reproducibility. *AJNR Am J Neuroradiol* 28,1949-1955 (2007).
2. Maldaner, N., et al. Interrater agreement in the radiologic characterization of ruptured intracranial aneurysms based on computed tomography angiography. *World Neurosurg* 103,876-882 (2017).
3. Bechan, R.S., et al. CT angiography versus 3D rotational angiography in patients with subarachnoid hemorrhage. *Neuroradiology* 57,1239–1246 (2015).
4. Han, J.F., Luo, P., Wang, X.G. Deep Self-Learning From Noisy Labels. *arXiv* 1908.02160 (2019).
5. Xiao, T., Xia, T., Yang, Y., Huang, C., Wang, X.G. Learning from massive noisy labeled data for image classification. In Proceedings of the IEEE Conference on Computer Vision and Pattern Recognition, pages 2691– 2699 (2015).

R3-5. The class balance is quite different between the different datasets used. In the training set 869/1177 (74%) patients had IAs. In the tuning/test set, the prevalence of aneurysms is 50%. In the NBH cohort, 39/211 (18%) patients had IAs. In Internal cohort 2, the number IAs is not mentioned. In Internal cohort 3, 47/151 (31%) patients had IAs. Prevalence in Internal cohort 5 is much lower at 10/214 (~5%), as DSA was not a requirement in these patients. No probability calibration is performed to correct for this.

Response:

Thank you for your comments. The prevalence of the aneurysms in each cohort is shown

in **Table R5** (also see **Table 1**) in the original version. Because we enrolled the consecutive cases in different clinical scenarios and different hospitals, the prevalence of the aneurysms in each cohort cannot be guaranteed by the researchers. We exposed our model in the real-world data to test whether the model can handle images from different hospitals with different protocols. So probability calibration in this study seems inappropriate.

Table R5. The prevalence of the aneurysms in each cohort.

	Internal cohort 1, n=1177	Internal cohort 2, n=245	Internal cohort 3, n=151	Internal cohort 4, n=374	Internal cohort 5, n=214	NBH cohort, n=211	TJ cohort, n=59	LYG cohort, n=316
Patients with IAs, n (%)	869 (73.8)	111 (45.3)	46 (30.5)	53 (14.2)	10 (4.7)	39 (18.5)	39 (66.1)	60 (19.0)
Number of IAs, n	1099	148	59	71	12	46	50	76
Patients without IAs, n (%)	308 (26.2)	134 (54.7)	105 (69.5)	321 (85.8)	204 (95.3)	172 (81.5)	20 (33.9)	256 (81.0)

R3-6. In independent test sets (Internal cohort 2) sensitivity was substantially poorer than in the test set (88.3% vs. 97.3%) and even worse in the NBH cohort (82.1%) or in Internal cohort 5 (40%). This indicates poor generalizability of the method to new datasets.

Response:

Thanks for your comments. We acknowledged this issue. We carefully double-checked the misdiagnosed cases. As we had discussed in R3-3, we found that some CTA images had inappropriate window width of 900 Hu and window level of 450 Hu [450,900] (**Fig. R2.**). So another two intervals containing [0, 450] and [-50, 650] were used to normalize the source images, and the performance of the model has increased a lot (**Table R3.**).

For the reasons why the sensitivity of the method decreased in some cohorts are as following: firstly, the prevalence of patients with aneurysms was quite lower in Internal cohort 3 (prevalence of 27.0%), 4 (14.2%) and 5 (4.2%), as well as LYG cohort (19.0%), which means the performances can be easily to decrease even though only few aneurysms misdiagnosed by the model. Secondly, the data in the training set was generated by Siemens Somatom Definition Flash or Somatom Definition with slice thickness of 0.5 mm. While the validation datasets from other hospitals are with different manufacturers, such as the images in the NBH cohort were generated by GE Optima 660 and SIEMENS SOMATOM Definition AS+ and the slice thickness is 0.625 mm and 0.6 mm, respectively. TJ cohort had multiple manufacturers (SIEMENS, GE and Toshiba). Our results demonstrated that different manufacturers had a great impact on the model performances, in which the model had a high performance in SIEMENS-generated CTA while lower in GE or Toshiba.

Internal cohort 5 enrolled patients who were suspicious for acute ischemic stroke (AIS). In this cohort, we aimed to study whether framework can work well when excluding the control cases to reduce workload in AIS setting, for that we had demonstrated that our framework had higher recall rate, sensitivity, and negative predictive value (NPV) than the radiologists, which may have the potential for complementary implementation in clinical practice. Therefore, with the triage of the framework, 87.4% of patients were predicted as negative, among which 96.8% predicted-negative cases are true-negatives, and the other 12.6% were predicted as high-risk group with the aneurysm. Therefore, radiologists can focus on these patients with more intense attention in order to improve workflow and reduce workload. In the revised manuscript, we have added cases and the results are encouraging (99.0% of predicted-negative cases are true-negatives), which indicates the potential for clinical application in real-world environment. On the other hand, in Internal cohort 4 and LYG cohort, our model demonstrated slightly higher sensitivity and recall rate than those of the radiologists and neurosurgeons (please see **Fig.2**).

R3-7. Throughout the paper, the authors list many values for sensitivity, specificity, recall, negative predictive value, accuracy, false positives/case. These are all quite connected and could be easily visualized in either a precision-recall plot, an ROC-curve, or an FROC-curve. For example, Fig.2 should be replaced with ROC curves that allow the reader to assess the differences between the automatic algorithm and the human observers. E.g., in Lines 294-296 the authors emphasize how their method has better sensitivity than humans. However, Fig. 2 shows that specificity and accuracy are lower than for humans. By picking a different operating point on the ROC curve, you can always find some criterion (sensitivity or specificity) on which you outperform the human, but what does that really mean?

Response:

We highly appreciate your suggestion. In this study, our model was based on the segmentation result, instead of predicting the probability of the existing of intracranial aneurysms directly. Our segmentation-based strategy allows us to achieve the location and shape of IAs, and the overlapping of the segmentation result and the ground truth lesion location was used to derive the detection results. With this strategy, a human-computer interaction interface can be achieved for radiologists' manipulation. However, the ROC/fROC curve is suitable for the classifier that output a probability, and they are used to analyze the performance of a classifier with different cut-off probabilities on the ROC/fROC curve. So the ROC/fROC curve may not suitable for our situation.

R3-8. In Lines 312-319 and Fig. 3 the authors describe how in Internal cohort 5, the method identified 12.6% of patients as containing an aneurysm, and how this is a good result as the radiologists do not have to focus on the other 87.4% of patients. However, in this cohort, only 5% of patients had aneurysms anyway, and the sensitivity is only 40%. Hence, 60% of aneurysms were missed by the algorithm and thus by a radiologist relying on this algorithm. This is a very poor result in this consecutive cohort.

Response:

Thanks for your comments. As we discussed in R3-3 and R3-6, the framework had a

higher performance after we modified the detection strategy. In the revised manuscript, the sensitivity of the model was 78.6% and specificity 89.7%. In the predicted-negative cases, 99.0% of predicted-negative cases are true-negatives.

In this study, we wondered whether framework can work well when excluding the control cases to reduce workload in acute ischemic stroke (AIS) setting, for that we had demonstrated that our framework had likely higher recall rate, sensitivity, and negative predictive value (NPV) than the radiologists, which may have the potential for complementary implementation in clinical practice. And patients who were suspicious for AIS had a quite low prevalence of aneurysm (4.2%). Therefore, with the triage of the framework, 86.8% of patients were predicted as negative, among which 99.0% predicted-negative cases are true-negatives, and the other 13.2% were predicted as high-risk group with the aneurysm (prevalence of 25.0%). Therefore, radiologists can focus on these patients with more intense attention in order to improve workflow and reduce workload.

For this population suspicious of AIS, the radiologists can rely on the algorithm to exclude patients without aneurysms rather than to detect patients with aneurysms one by one, a very low efficiency. Radiologists are required to focus on the high risk patients with more attention, in which situation the algorithm acts as an alert.

R3-9. Line 170: In the NBH cohort, the method misses IAs in one in three patients.

Response:

In the initial version, 13 aneurysms from 13 patients (including 6 IAs in 6 patients with multiple IAs) were missed in the revised version. After carefully checked the data, 1 of the missed aneurysm can be categorized to occult aneurysm, so it is excluded from the cohort during the revision. Besides, we have two metrics to describe our results, which are patient-based indices (accuracy, sensitivity, specificity, PPV, and NPV) and aneurysm-based indices (recall rate, FP/case and Dice coefficient). Therefore, in the NBH cohort, 12 aneurysms from 12 patients were missed, among which 7 patients had only one aneurysm and the 7 patients can be treated as missed patients, that is the method missed IAs in one in five patients (7/39). And 34 aneurysms were detected among the 46 ground truths based on the aneurysm-based analysis (recall rate of 73.9%), with Dice coefficient of 0.510.

In the revised manuscript, the model's performance has increased a lot after we modified the model. 11 aneurysms from 6 patients (including 5 IAs in 5 patients with multiple IAs) were missed among 39 patients containing 46 aneurysms. So the sensitivity and recall rate are 84.6% and 76.1%, respectively.

R3-10. Lines 237-249. Were these patients from the 'general population', for whom DSA images were not always available?

Response:

As we have mentioned in the Data section in the Online methods part, cases used for clinical application in routine practice and comparison with radiologists were consecutive patients with suspected intracranial aneurysms or other cerebral vascular diseases. Therefore, these patients were not the 'general population', which often refers to the

ordinary individuals and used in the epidemiological investigation and community research.

R3-11. It would be good to include more images of cases in which the method worked or did not work well.

Response:

Highly appreciate your suggestion. We have added another series of cases that the method worked well or did not work well.

Besides, we have enrolled the consecutive cases in additional temporally or spatially independent data sets for further validation. The added data were shown in **Table R4**.

R3-12. Lines 130-131: It's unclear what was segmented in the images.

Response:

The aneurysms' sac was segmented in the images, which acted as the ground truth (**Fig. R4**). The overlapping of the segmentation result and the ground truth lesion location was used to derive the detection results based on patient and aneurysm (**Fig. R5, please also see Extended Data Fig.1**).

Fig. R4. Sketch map for the segmented process.

Fig. R5. Overview of the row image, ground truth and the prediction result. This is the segmentation-based methods used to achieve the detection task. The first picture is a cross-sectional image of bone-removal CT image of a patient with right MCA aneurysm. The middle picture shows the segmented result (red circle). The third picture shows how the predicted results presented.

R3-13. Line 127-128: Why do the authors mention the total number of 2D slices? This is irrelevant as the method operates on 3D image patches.

Response:

Revised as suggested.

R3-14. Line 140: Number of FPs is set to be 0.3, but what does that mean? 0.3 FP per CTA scan?

Response:

Revised as suggested.

R3-15. Lines 156-157: What do the authors consider to be the difference between sensitivity and recall rate? Aren't these the same?

Response:

In our study, the sensitivity is for patient-level while the recall rate is for lesion-level. As there may exist more than one IAs in some CTA scans, both patient-level and lesion-level recalls are given to verify the performance comprehensively.

R3-16. Lines 177-178: Why are there three p values for two cohorts? What kind of statistical test was used here?

Response:

As we have mentioned in the Statistical analysis section in the Online methods part, "To assess model performance against that those of 6 radiologists, we used a 2-sided Pearson's chi-squared test to evaluate whether there were significant differences in specificity, sensitivity, accuracy, PPV, and NPV between the framework and radiologists." The analysis was conducted in the whole group, the SAH group and the non-SAH group, so there are three *p* values for the two cohorts (please see the footnote of **Extended Data Table 3** in the Supplementary file).

R3-17. Lines 200-203: I don't really see a pattern between image quality and model performance. Was there any?

Response:

Thanks for your comments. No statistically significant differences were found for the performance of the model among different image qualities of CTA, please see the 8th sentence in page 10.

R3-18. Lines 206-207. Barely is a bit exaggeration. 10/151 is still around 6%.

Response:

Thanks for your comments. We have revised the sentence "We can barely collect 10 cases among all head CTA cases, which means the model is qualified for further application" as to "We only collected 11 cases with poor image quality among all head CTA cases, which meant the model requires further validation."

R3-19. Line 250: What is meant with microaverage sensitivity?

Response:

As we have mentioned in the Statistical analysis section in the Online methods part, “*The micro-average of sensitivity, specificity, accuracy, PPV and NPV across all radiologists were computed by measuring each statistic pertaining to the total number of true-positive, true-negative, false-positive and false-negative results.*”

R3-20. - Typos, grammar, e.g.

o Line 145. Comma after ‘That is’

o Line 211: patents > patients

o Line 258: NVP > NPV

o Line 293: AIS > IAS

o Line 452: running > tuning

Response:

Revised as suggested.

Reviewers' Comments:

Reviewer #2:

Remarks to the Author:

Thanks to the authors for improving the manuscript.

Some minor comments:

R2-3.: the reference quoted in regard to the AI chasm is not relevant, and the link does not refer to it. Please quote a proper reference

R2-5: My previous comment has not been addressed. The sentence is still broken, and CTA is used to see vessels in general, not "the human brain"

"Administration of antiplatelet or anticoagulant therapy is often recommended, which will increase risk of aneurysm rupture¹³". This is controversial, and more relevant references should support it

"It is just an experience value and the number of training epochs to 100 is frequently used by other researchers^{1,2}. ". The references adds into the comments should be added into the manuscript. And the references should be checked for relevance, as many times I see papers quoted in the wrong way (for example, they quote some papers in regard to something else, not quoting the source or most relevant reference).

Reviewer #3:

Remarks to the Author:

The authors have carefully considered my comments and have justified their decisions well. The manuscript has been greatly improved and additional experiments have been included. I have some small comments remaining:

- I do think it's still confusing that the authors use 'recall' and 'sensitivity' interchangeably, and would suggest to use e.g. 'patient-level sensitivity' and 'lesion-level sensitivity'.
- How is it that the window and level are relevant for the CNN? Do the authors pre-process/clip the CTA images in some way? I couldn't find that in the method description. If not, shouldn't the CNN be able to automatically determine the right window-level?

Department of Diagnostic Radiology
Jinling Hospital, Medical School of Nanjing University
305 Zhongshan East Road, Nanjing
Jiangsu Province, 210002, China
Email: kevinzhlj@163.com

October 03, 2020

Dear referees,

We would like to give a lot of thanks to the referees for your helpful and insightful comments for our manuscript (**NCOMMS-20-10696A**) entitled “Clinically Applicable Deep Learning for Intracranial Aneurysm Detection in Computed Tomography Angiography Images: A Comprehensive Multicohort Study”, which have significantly improved the manuscript. We have taken all comments seriously and carefully revised the manuscript according to the suggestions.

Our detailed responses to the specific comments are presented in the next pages. The original comments are in *a red italic font*, and our responses are in a black regular font.

Again, thank you very much and look forward to hearing from you soon!

Sincerely yours,

Long Jiang Zhang, M.D., Ph.D.,

Reviewer #2 (Remarks to the Author):

Thanks to the authors for improving the manuscript.

Some minor comments:

Response:

Thank you for your positive comments and encouragement.

R2-3: the reference quoted in regard to the AI chasm is not relevant, and the link does not refer to it. Please quote a proper reference

Response:

Thanks for your suggestion. As suggested, we have replaced the reference with another one which had elaborated the issue of *AI chasm*¹.

Reference

1. Keane, P. & Topol, E. With an eye to AI and autonomous diagnosis. *NPJ Digit Med* **1**, 40 (2018).

R2-5: My previous comment has not been addressed. The sentence is still broken, and CTA is used to see vessels in general, not "the human brain"

Response:

Thank you for your comments. We found that this sentence might cause some confusion, therefore we removed it and revised the sentence as "Therefore, we collected 1,177 digital subtraction head bone-removal CTA images, which were based on a section-by-section subtraction to subtract nonenhanced from enhanced CT data to facilitate the diagnosis of aneurysms, with/without SAH to derive a specific model for automated detection of IAs".

R2-6. "Administration of antiplatelet or anticoagulant therapy is often recommended, which will increase risk of aneurysm rupture¹³". This is controversial, and more relevant references should support it.

Response:

Thank you for your comments. We have examined the literature and discussed this issue with neurosurgeons. We acknowledged whether the risk of bleeding from unruptured intracranial aneurysms increases during the treatment of acute ischemic stroke (AIS) remains controversial. Several researchers have reported the catastrophic intracranial aneurysmal rupture following intravenous thrombolytic therapy in these patients^{1,2}. We revised the sentence as "Intravenous thrombolysis is efficacious and safe for AIS patients, while it might increase risk of aneurysm rupture in some reports^{1,2}".

References

1. Haji, F., van, Adel, B., Avery, M., Megyesi, J., Young, G.B. Intracranial aneurysm rupture following intravenous thrombolysis for stroke. *Can J Neurol Sci* **41**,95-98 (2014).
2. Zaldivar-Jolissaint, J.F., Messerer, M., Bervini, D., Mosimann, P.J., Levivier, M., Daniel, R.T. Rupture of a concealed aneurysm after intravenous thrombolysis of a thrombus in the parent middle cerebral artery. *J Stroke Cerebrovasc Dis* **24**, e63–e65 (2015).

R2-7. *"It is just an experience value and the number of training epochs to 100 is frequently used by other researchers^{1,2}. ". The references adds into the comments should be added into the manuscript. And the references should be checked for relevance, as many times I see papers quoted in the wrong way (for example, they quote some papers in regard to something else, not quoting the source or most relevant reference).*

Response:

Thank you for your suggestion. We have double checked the references for relevance and the network was trained for 100 epochs¹. We added it into the manuscript as suggested. We also double checked all references in this revised manuscript.

References

1. Mehta, S., et al. Y-Net: Joint segmentation and classification for diagnosis of breast biopsy images. *arXiv* 1806.01313 (2018).

Reviewer #3 (Remarks to the Author):

R3-1. *The authors have carefully considered my comments and have justified their decisions well. The manuscript has been greatly improved and additional experiments have been included. I have some small comments remaining.*

Response:

Thank you for your excellent comments and your suggestions to improve this manuscript.

R3-2. - *I do think it's still confusing that the authors use 'recall' and 'sensitivity' interchangeably, and would suggest to use e.g. 'patient-level sensitivity' and 'lesion-level sensitivity'.*

Response:

Highly appreciate your kind suggestion. Revised as suggested.

R3-3. - *How is it that the window and level are relevant for the CNN? Do the authors pre-process/clip the CTA images in some way? I couldn't find that in the method description. If not, shouldn't the CNN be able to automatically determine the right window-level?*

Response:

Thank you for your excellent comments. We carefully double checked the misdiagnosed cases and found that some CTA images had inappropriate brightness under the default interval [0, 900] (window width of 900 Hounsfield units (Hu) and window level of 450 Hu), which made the images too dark to detect the vessels and aneurysms. After adjusting the two intervals of [0, 450] and [-50, 650] (window width and window level of [300,700] and [225,450], respectively), the images become brighter and the aneurysms are obviously shown, and the aneurysms can be recalled by the model (**Fig. R1**).

Fig. R1. Examples of the misdiagnosed cases with default interval of $[0, 900]$ and the adjusting window interval ($[0, 450]$ and $[-50, 650]$). **Panel a**, patients with an aneurysm in the anterior communication artery and posterior communication artery (arrows), respectively. The first picture shows CTA-derived volume rendering (VR) image with an aneurysm (arrow); the second shows the raw cross-sectional digital subtraction bone-removal CT image with the default window interval of $[0, 900]$; the third shows the ground truth of aneurysm (red outline in the box); the fourth shows the rightly predicted result following adjusting window interval of $[300, 700]$ (red outline in the box), and the fifth picture is an zoomed image of the predicted aneurysm (red outline). **Panel b**, patients with an aneurysm in the anterior communication artery and vertebral basilar artery (arrows), respectively. The aneurysms were missed in the default window interval of $[0, 900]$ while they are correctly diagnosed following adjusting window interval of $[0, 450]$. After adjusting the window interval to $[-50, 650]$ and $[0, 450]$, the images become brighter and the aneurysms are obviously shown, and the aneurysms can be recalled by the model.

In order to detect IAs in some low-contrast images clipped by the default interval $[0, 900]$, another two intervals of $[0, 450]$ and $[-50, 650]$ were used to normalize the source images. The setting was automatically selected according to the brightness distribution. Given a bone removal CTA image, a threshold value such as 150 Hu was used to find the initial area of vessels and then the maximum connectivity area was kept as the final region of vessels. Histogram of the brightness of voxels in the region was analyzed to find suitable clipping interval. If the brightness is mainly concentrated above 300 Hu, the default

clipping interval was used. Otherwise, we counted the distribution of two intervals including $[0, 200]$ and $[200, 300]$, which corresponded to clipping intervals of $[0, 450]$, $[-50, 650]$, respectively (**Fig. R2**), and the clipping interval corresponding to the dominated distribution interval was selected to normalize the source image. We have added it in the method description in the revised manuscript (see the red highlight in the revision).

Fig. R2. Histograms of the brightness distribution from three CTA images. **Panel a**, the dominated interval is $[0, 200]$, the clipping interval of $[0, 450]$ is selected. **Panel b**, the dominated interval is $[200, 300]$, the clipping interval of $[-50, 650]$ is selected. **Panel c**, the brightness is mainly concentrated above 300 Hu, the clipping interval of $[0, 900]$ is selected.